# Chemical genetics strategy to profile kinase target engagement reveals role of FES in neutrophil phagocytosis

Tom van der Wel [1], Riet Hilhorst [2], Hans den Dulk[1], Tim van den Hooven[2], Nienke M. Prins[1], Joost A. P. M. Wijnakker[1], Bogdan I. Florea[3], Eelke B. Lenselink[4], Gerard J. P. van Westen [4], Rob Ruijtenbeek[2], Herman S. Overkleeft[3], Allard Kaptein[5], Tjeerd Barf[5] & Mario van der Stelt [1]✉

Chemical tools to monitor drug-target engagement of endogenously expressed protein kinases are highly desirable for preclinical target validation in drug discovery. Here, we describe a chemical genetics strategy to selectively study target engagement of endogenous kinases. By substituting a serine residue into cysteine at the DFG-1 position in the ATP-binding pocket, we sensitize the non-receptor tyrosine kinase FES towards covalent labeling by a complementary fluorescent chemical probe. This mutation is introduced in the endogenous *FES* gene of HL-60 cells using CRISPR/Cas9 gene editing. Leveraging the temporal and acute control offered by our strategy, we show that FES activity is dispensable for differentiation of HL-60 cells towards macrophages. Instead, FES plays a key role in neutrophil phagocytosis via SYK kinase activation. This chemical genetics strategy holds promise as a target validation method for kinases.

---

[1] Department of Molecular Physiology, Leiden Institute of Chemistry, Leiden University & Oncode Institute, Leiden, The Netherlands. [2] PamGene International BV, 's-Hertogenbosch, The Netherlands. [3] Department of Bio-organic Synthesis, Leiden Institute of Chemistry, Leiden University, Leiden, The Netherlands. [4] Department of Drug Discovery & Safety, Leiden Academic Centre for Drug Research, Leiden University, Leiden, The Netherlands. [5] Covalation Biosciences BV, Ravenstein, The Netherlands. ✉email: m.van.der.stelt@chem.leidenuniv.nl

Protein kinases comprise a 518-membered family of enzymes that play essential roles in intracellular signaling processes. They transfer a phosphate group from ATP to specific amino acid residues in proteins, thereby modulating protein activity, localization and protein–protein interactions[1,2]. Protein kinases are involved in many cellular functions, including proliferation, differentiation, migration, and host–pathogen interactions. Kinases are also an important class of drug targets for the treatment of cancer[3]. However, current FDA-approved kinase inhibitors are designed to target only <5% of the entire kinome[4] and therapeutic indications outside oncology are vastly underrepresented[5,6]. These so-far untargeted kinases thus offer great opportunities for the development of novel molecular therapies for various diseases. The non-receptor tyrosine kinase feline sarcoma oncogene (FES), subject of the here presented study, is a potential therapeutic target for cancer and immune disorders[7–9].

FES, together with FES-related kinase (FER), constitutes a distinct subgroup within the family of tyrosine kinases, defined by their unique structural organization. FES is able to form oligomers via its F-Bin-Amphiphysin-Rvs (F-BAR) domain, which drives translocation from the cytosol to the cell membrane[10,11]. In addition, FES possesses a Src Homology 2 (SH2) domain that binds phosphorylated tyrosine residues and functions as protein interaction domain[12,13]. The catalytic domain of FES is located on its C-terminal end. Phosphorylation of Y713 in the activation loop of FES is a prerequisite for its kinase activity and can occur either via autophosphorylation or phosphorylation by Src family kinases[14,15]. FES has a restricted expression pattern and is found in neuronal, endothelial and epithelial cells. Its expression levels are highest in cells of hematopoietic origin, especially those in the myeloid lineage[16]. Most of the physiological processes of FES have been studied in macrophages[17,18] or mast cells[19], but far less is known about its role in other terminally differentiated myeloid cells, such as neutrophils.

The successful development of new kinase-targeting drugs strongly depends on our understanding of their underlying molecular and cellular mechanism of action, i.e. the preclinical target validation[20]. The physiological function of many kinases remains, however, poorly characterized and their direct protein substrates are often unknown. Genetic models (congenital deletion or expression of kinase-dead variants) may be used to study these questions. For example, FES knockout mice revealed a role for FES in leukocyte migration[21,22] and the release of inflammatory mediators[17]. However, long-term, constitutive genetic disruption of kinases can result in compensatory mechanisms that counteract defects in cellular signaling. For example, mice lacking both FES and FER have more pronounced defects in hematopoiesis than counterparts lacking only one of these kinases, which may indicate that these related kinases may compensate for each other's loss[23]. Long-term, permanent genetic models are therefore poorly suited to study rapid and dynamic signaling processes[24,25]. In addition, phenotypic differences between independently generated knockout animals are not uncommon, as was the case for two independently developed *fes*[−/−] mice[17,26]. A complementary approach is the use of inhibitors to modulate kinase activity in an acute and temporal fashion. This approach more closely resembles therapeutic intervention, but the available pharmacological tools, especially for non-validated kinases, often suffer from a lack of selectivity[24,27]. Currently, there are no suitable FES inhibitors available for target validation studies, because they either lack potency or selectivity, and all cross-react with FER[7,28].

A key step in the target validation process consists of obtaining proof of target engagement, which is essential to correlate inhibitor exposure at the site of action to a pharmacological and phenotypic readout[29]. Information about kinase engagement is also useful for determining the dose required for full target occupancy without inducing undesired off-target activity[30]. Chemical probes that make use of a covalent, irreversible mode of action are ideally suited to study target engagement[29]. Incorporation of reporter tags enable target visualization (e.g. fluorophores) or target enrichment and identification (e.g. biotin). In the field of kinases, reported chemical probes either target a conserved active-site lysine residue in a non-selective fashion[31] or non-catalytic cysteine residues in the ATP-binding pocket[32,33]. The first class of kinase probes lacks the selectivity required for cellular target engagement studies. On the other hand, the majority of kinases, including FES, do not possess targetable cysteine residues in the catalytic pocket[34].

Garske et al. previously introduced the elegant concept of covalent complementarity: the use of an engineered kinase in which the gatekeeper amino acid residue is mutated into a cysteine, combined with electrophilic ATP analogs to study target engagement[35]. Other positions in the kinase active site have also been investigated[36–38], but secondary mutations were required to improve cysteine reactivity or compound selectivity and potency. Of note, all studies relied on transient or stable overexpression of the mutant kinase, rather than physiological model systems with endogenous expression levels. Since overexpression of kinases is known to disrupt physiological intracellular signaling cascades[39], there is a need for target validation methods that visualize specific endogenous kinase activity and its engagement by small molecules without perturbing normal cellular processes. Inspired by these established and emerging concepts, we describe herein a chemical genetics strategy to profile acute target engagement of FES kinase by a complementary, mutant-specific chemical probe.

## Results

**Chemical genetics strategy for target engagement studies**. The key feature of the chemical genetics strategy is the combination of an engineered, mutant kinase with the design and application of complementary, covalent inhibitors (Fig. 1a). The ATP-binding pocket of the kinase of interest is sensitized towards pharmacological inactivation using complementary probes by substituting an amino acid for a cysteine residue. The mutant is biochemically characterized to verify that this mutation minimally affects kinase function (Fig. 1b, step 1). Subsequently, complementary electrophilic inhibitors are designed to covalently react with this cysteine (Fig. 1b, step 2). Covalent, irreversible inhibitors can have several advantages over reversible compounds, such as sustained target occupancy, lower susceptibility to competition by high intracellular ATP concentrations and a pharmacodynamic profile that is dependent on the target's de novo protein synthesis rate[40]. The inhibitor will have far lower potency on the wild-type (WT) kinase, which does not possess a nucleophilic residue in its active site, thereby making the inhibitor mutant-specific. An important feature of our strategy is that the corresponding cysteine point mutation is introduced endogenously in a relevant cell line using CRISPR/Cas9 gene editing (Fig. 1b, step 3). This circumvents the use of cellular overexpression systems, which may disturb the intrinsic balance of the kinase signaling network, leading to compensatory effects and other artefacts in cellular function[39]. Importantly, the covalent binding mode of the inhibitor enables target engagement profiling by acting as a chemical probe and its ligation handle can be further functionalized with reporter tags for visualization by SDS-PAGE (fluorophore) or identification of the bound targets by mass spectrometry (biotin) (Fig. 1b, step 4). Since this approach allows acute inactivation of kinases with high specificity, it can be employed to study kinase function and thereby aid in its validation as therapeutic target (Fig. 1b, step 5).

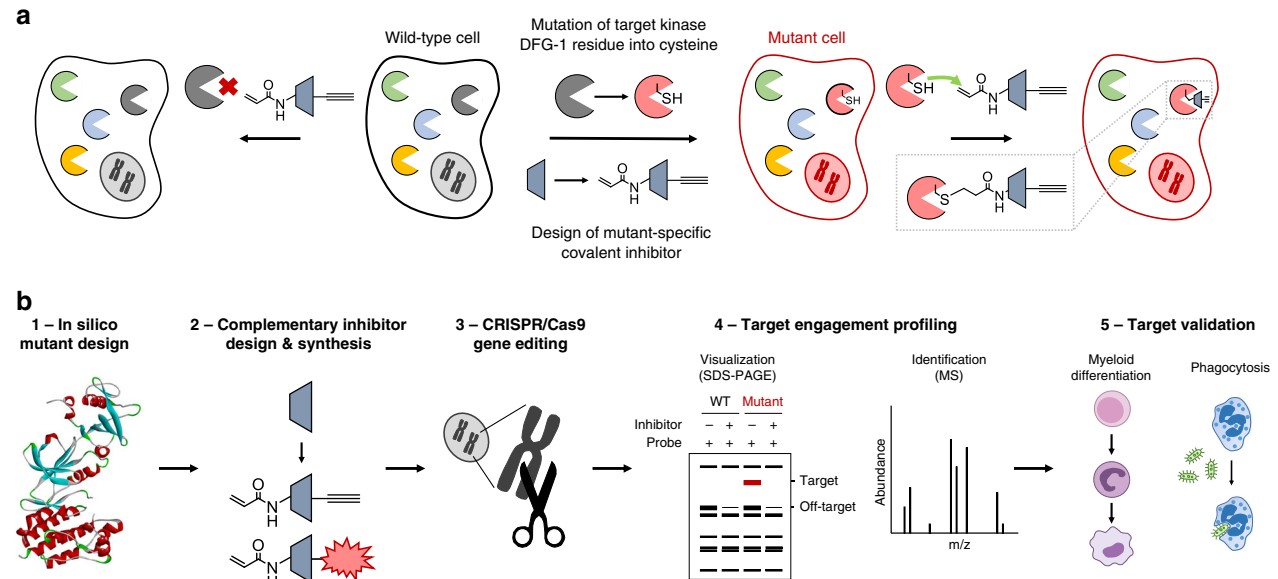

**Fig. 1 Chemical genetics strategy to visualize kinase activity and target engagement. a** General strategy involving mutagenesis of a kinase DFG-1 residue into a nucleophilic cysteine accompanied by mutant-specific covalent inhibitor design. A non-selective, reversible inhibitor is modified with an acrylamide electrophile to covalently react to the introduced cysteine, along with an alkyne ligation handle for (bioorthogonal) conjugation to reporter tags using click chemistry. The mutation is introduced in a relevant cell line using CRISPR/Cas9 gene editing in order to study the kinase in an endogenous expression system. **b** Schematic workflow for application of mutant-specific, complementary probes in target engagement and target validation studies.

**Biochemical characterization of engineered FES kinases.** To introduce a cysteine residue at an appropriate position in the ATP-binding pocket of FES, we inspected the reported crystal structure of FES with reversible inhibitor TAE684 (compound **1**) (PDB: 4e93)[28]. We selected nine active-site residues situated in proximity of the bound ligand (Fig. 2a) and generated the respective cysteine point mutants by site-directed mutagenesis on truncated human FES (SH2 and kinase domain, residues 448–822) fused to a N-terminal His-tag. The WT protein and the mutants were recombinantly expressed in *Escherichia coli* (*E. coli*), purified using $Ni^{2+}$-affinity chromatography and tested for catalytic activity using a time-resolved fluorescence resonance energy transfer (TR-FRET) assay (Fig. 2b). Four of the nine tested mutants did not display any catalytic activity, including G570C (located on P-loop) and G642C (hinge region). Three mutants near the kinase hydrophobic backpocket (I567C, V575C, and L638C) retained partial activity, whereas only two mutants (T646C and S700C) displayed catalytic activity similar to FES$^{WT}$. Our attention was particularly drawn to the S700C mutant, which involves the residue adjacent to the highly conserved DFG motif (DFG-1). Since several other kinases (e.g. MAPK1/3, RSK1-4, and TAK1) express an endogenous cysteine at DFG-1 that can be targeted by electrophilic traps[32], we chose to profile FES$^{S700C}$ in more detail. The engineered kinase displayed identical reaction progress kinetics (Fig. 2c) and similar affinity for ATP ($K_M = 1.9$ μM for FES$^{WT}$ and $K_M = 0.79$ μM for FES$^{S700C}$; Fig. 2d and Supplementary Fig. 1a).

To assess whether the introduced mutation affected substrate recognition, a comparative substrate profiling assay was performed using the PamChip® microarray technology. This assay is based on the phosphorylation of immobilized peptides by purified FES and detection using a fluorescently labeled anti-phosphotyrosine antibody. Strikingly, the substrate profiles of FES$^{WT}$ and FES$^{S700C}$ were completely identical (Fig. 2e, inset; Supplementary Table 1), indicating that the S700C mutation did not affect substrate recognition. Moreover, the absolute peptide phosphorylation levels showed a strong correlation ($R^2 = 0.95$). Comparison with the substrate profiles of five other non-receptor

tyrosine kinases (ABL, CSK, FGR, LYN, and SYK) confirmed that a substantial number of peptides are FES-specific substrates (Supplementary Fig. 2). Sequence analysis of the top 30 of highest signal peptides revealed that FES prefers negatively charged substrates with hydrophobic residues at positions −1 and +3 and acidic residues at position −4, −3, and +1 relative to the tyrosine phosphorylation site (Fig. 2f). These results are in line with a previous study that reported on FES substrate recognition[41]. Lastly, a modified PamChip array to measure phosphopeptide binding to the Src homology 2 (SH2) domain of FES$^{WT}$ and FES$^{S700C}$ showed that the introduced mutation did not affect the SH2 binding profile (Fig. 2g and Supplementary Table 2). In short, the DFG-1 residue (Ser700) in the ATP-binding pocket of FES was identified as an excellent position to mutate into a nucleophilic cysteine, without affecting FES kinase activity, kinetics, substrate recognition or SH2 binding profile.

**Synthesis and characterization of complementary probes.** The reversible ligand TAE684 was used as starting point to develop a complementary probe for FES$^{S700C}$. To assess whether the FES$^{S700C}$ was still sensitive to inhibition by TAE684, the protein was incubated with various concentrations of TAE684 and its half maximum inhibitory concentration (expressed as pIC$_{50}$) was determined. We found that TAE684 was a potent inhibitor both on FES$^{WT}$ and FES$^{S700C}$ with pIC$_{50}$ values of 8.1 ± 0.04 and 9.0 ± 0.02, respectively (Table 1; assay performed with 5 μM ATP). According to the co-crystal structure of FES$^{WT}$ with TAE684 (PDB: 4e93) the isopropyl sulfone moiety is in the close proximity of Ser700 at the DFG-1 position. Therefore, several derivatives of TAE684 were synthesized, in which the R$_2$-phenyl was substituted with an acrylamide group as electrophilic warhead (Supplementary Methods). The acrylamide is hypothesized to covalently interact with the engineered cysteine, but not with the serine of the WT protein. Since the strategy aims at exclusively inhibiting mutant but not WT FES, the piperidine-piperazine group was removed as it forms water-mediated hydrogen bonds with hinge region residues and contributes

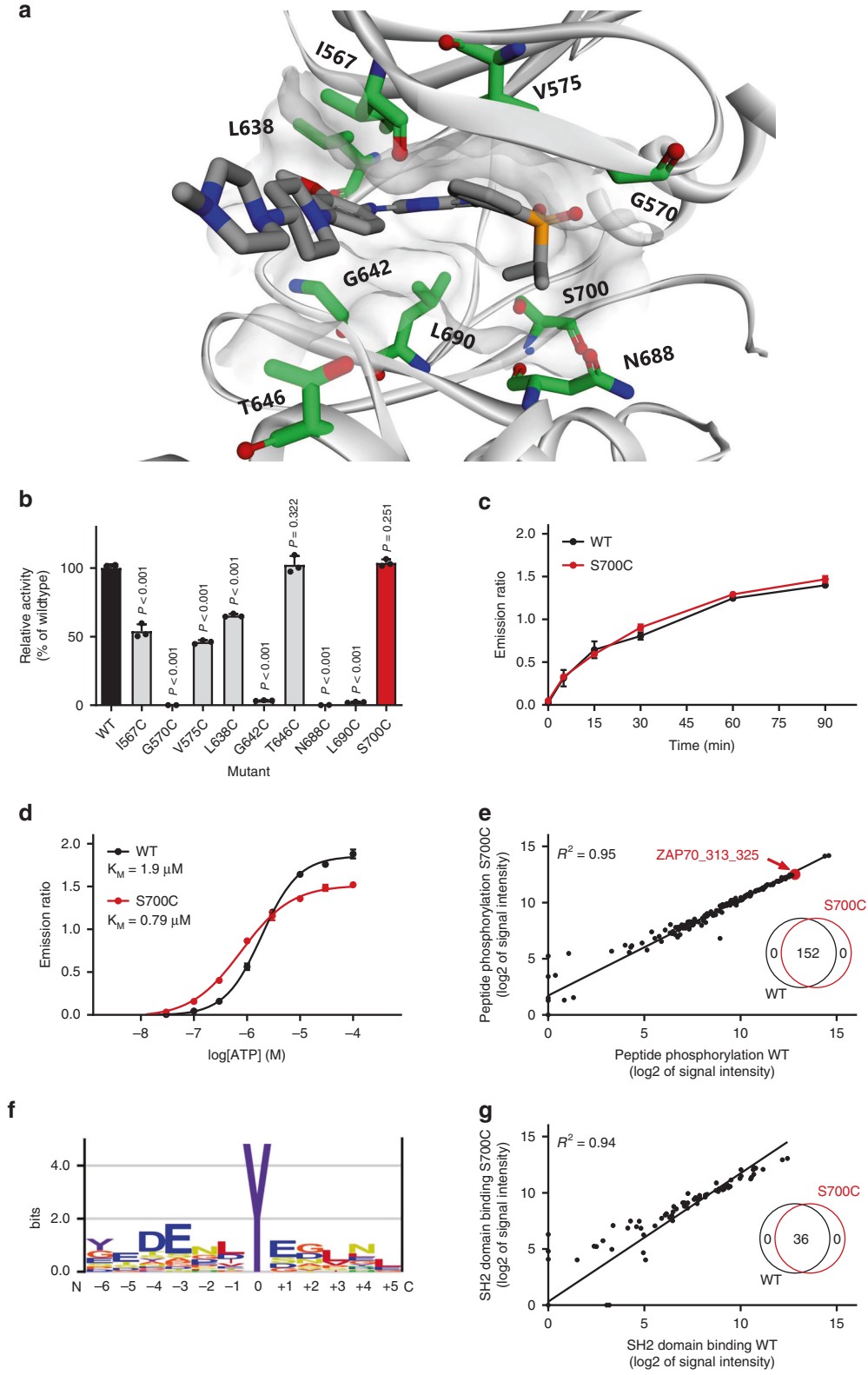

to ligand affinity[28]. The pIC$_{50}$ values of compound **2–6** are listed in Table 1 (dose-response curves in Supplementary Fig. 1,b–f). Removal of the piperidine-piperazine group (compound **2**) resulted in a modest reduction in potency on FES$^{WT}$ and FES$^{S700C}$. Introduction of an acrylamide at the *meta*-position of the phenyl ring (compound **3**) further decreased affinity for FES$^{WT}$, but also led to significant loss in activity on the FES$^{S700C}$ mutant. Moving the acrylamide to the *ortho*-position resulted in

compound **4**, which exhibited excellent potency on FES$^{S700C}$ (pIC$_{50}$ = 8.4 ± 0.03), whereas a major reduction in potency on FES$^{WT}$ (pIC$_{50}$ = 5.7 ± 0.21) was found, resulting in an apparent selectivity window of 238-fold. We substituted the acrylamide with a propionyl amide (compound **5**) to confirm its important role in binding to FES$^{S700C}$. In line with the proposed mode of action, compound **5** displayed low inhibitory potency on FES$^{S700C}$.

**Fig. 2 Design and characterization of FES cysteine point mutants. a** Location of mutated active-site residues (green) in FES crystal structure with bound reversible inhibitor TAE684 (PDB code: 4e93). **b** Activity of recombinantly expressed FES mutants compared to WT, determined as relative amount of phosphorylated peptide substrate after 60 min incubation using TR-FRET assay. **c** Reaction progress kinetics for $FES^{WT}$ and $FES^{S700C}$. **d** Determination of ATP $K_M$ for $FES^{WT}$ and $FES^{S700C}$. Enzyme reactions in TR-FRET assay were performed with *ULight*-TK peptide (50 nM) and ATP (**b**, **c**: 100 μM, **d**: variable) and quenched (**b**, **d**: after 60 min, **c**: variable). **e** Peptide phosphorylation substrate profile for $FES^{WT}$ and $FES^{S700C}$ as determined in PamChip® microarray. Peptides were filtered for those with ATP-dependent signal and log2 of signal intensity >3. The peptide substrates were identical for $FES^{WT}$ and $FES^{S700C}$ (Venn diagram, inset). **f** Preferred substrate consensus sequence based on $FES^{WT}$ substrate profile. Illustration was rendered using Enologos (http://www.benoslab.pitt.edu). **g** SH2 domain binding profile for $FES^{WT}$ and $FES^{S700C}$ as determined in PamChip® microarray. Peptides with non-specific antibody binding were excluded. The peptide SH2 binding partners were identical for $FES^{WT}$ and $FES^{S700C}$ (Venn diagram, inset). Data represent means ± SEM ($n = 3$). Statistical analysis: ANOVA with Holm–Sidak's multiple comparisons correction: *** $P < 0.001$; NS if $P > 0.05$. Source data are provided as a Source Data file.

Next, we performed a docking study with compound **4** in $FES^{S700C}$ (Fig. 3a). The binding mode of **4** resembled the original binding pose of TAE684 and could explain the observed structure-activity relationships. Catalytic lysine residue 590 interacts with the amide carbonyl, ideally positioning the warhead on the *ortho*-position, but not *meta*-position, to undergo a Michael addition with the engineered cysteine. The binding pose also revealed a suitable position to install an alkyne moiety on the scaffold of compound **4** to develop a two-step chemical probe for target engagement studies. This led to the synthesis of probe **6** (hereafter referred to as WEL028) with an *ortho*-acrylamide and alkoxyalkyne, which displayed a similar potency profile as **4** with strong inhibition of $FES^{S700C}$ ($pIC_{50} = 8.4 \pm 0.03$) but not $FES^{WT}$ ($pIC_{50} = 5.0 \pm 0.37$) (Fig. 3b). The mutant-specific inhibition profile was additionally verified using the orthogonal PamChip® microarray assay (Fig. 3c). To confirm that WEL028 undergoes Michael addition to cysteine 700, we analyzed the WEL028-$FES^{S700C}$ complex in more detail using mass spectrometry. Recombinant $FES^{S700C}$ was incubated with WEL028, digested to peptide fragments with trypsin and subsequently analyzed by LC-MS/MS to confirm covalent addition of WEL028 to Cys700 (Fig. 3d and Supplementary Fig. 3).

Cys700 is located directly adjacent to the highly conserved DFG motif. A substantial number of kinases also harbors a native cysteine at this position, which might have implications for the kinome-wide selectivity of WEL028[40]. We therefore assessed the selectivity using the SelectScreen™ screening technology in a panel of 380 wild-type, mammalian kinases including all kinases with a native cysteine residue at any position in the active site. The assays were performed at a single dose of 1 μM with 1 h of preincubation to identify the full spectrum of potential off-target kinases (Supplementary Fig. 4). Out of the 380 tested kinases, 26 showed >50% inhibition under these conditions, meaning that WEL028 exhibited a >100-fold selectivity window against the residual 354 kinases (93% of tested kinases, Fig. 3e). Subsequently, dose-response experiments were performed for a representative selection of kinases showing >50% inhibition in the initial screen (Supplementary Table 3 and Supplementary Fig. 5). In short, WEL028 was identified as a complementary two-step probe for engineered $FES^{S700C}$ that does not label $FES^{WT}$ or FER and has a sufficient selectivity profile over other kinases for our purposes.

**Target engagement studies with one-step probe WEL033.** Next, a one-step fluorescent probe for $FES^{S700C}$ was synthesized to facilitate visualization of target engagement: a Cy5-conjugated analog of WEL028 termed WEL033 (Fig. 4a). WEL033 dose-dependently labeled recombinantly expressed full-length $FES^{S700C}$ but not $FES^{WT}$ in HEK293T cell lysate (Fig. 4b). Similar results were obtained using two-step labeling of WEL028-treated lysate clicked to Cy5-azide in vitro (Supplementary Fig. 6a, b). Complete labeling was achieved within 15 min and this

labeling was stable up to 60 min (Supplementary Fig. 6c). Fluorescent labeling of $FES^{S700C}$ was observed regardless of its autophosphorylation state, suggesting that WEL033 covalently binds to both catalytically active and inactive FES (Supplementary Fig. 7a). Interestingly, introduction of a secondary K590E mutation abolished labeling by WEL033 (Fig. 4c). This could indicate that Lys590 is essential for covalent binding of WEL028 to the FES active site, possibly by coordination of Lys590 that positions the acrylamide warhead to undergo covalent addition to Cys700 as predicted by docking studies (Fig. 3a).

To visualize target engagement of $FES^{S700C}$, a competitive binding assay was performed with WEL033 in lysates over-expressing full-length $FES^{S700C}$. Two inhibitors (TAE684 and WEL028) were able to prevent the labeling of $FES^{S700C}$ in a dose-dependent manner ($pIC_{50} = 7.6 \pm 0.06$ and $pIC_{50} = 7.9 \pm 0.06$, respectively; Fig. 4d, e). Of note, we found that our complementary probes could also be used to visualize target engagement on other kinases with the DFG-1 residue mutated into a cysteine, such as $FER^{S701C}$, $LYN^{A384C}$, $PTK2^{G536C}$, and $PAK4^{S457C}$ (Supplementary Figs. 8 and 9, Supplementary Table 4 and Supplementary Note 1).

Time-dependent displacement experiments were performed to study the mode of action of TAE684 and WEL028 (Supplementary Fig. 7b, c). After inhibitor incubation at their respective $IC_{80}$ concentrations, labeling of $FES^{S700C}$ activity by WEL033 recovers for TAE684 but not for WEL028, indicating a reversible and irreversible mode of action for these compounds, respectively. In line with these results, inhibitor washout experiments using dialysis also showed sustained inhibition by WEL028 but not TAE684 (Supplementary Fig. 7d–g). Of note, the increased labeling intensities with TAE684-treated protein might be due to enhanced stability of the protein as commonly observed for molecular chaperones[42]. Thus, gel-based probe labeling experiments using lysates of cells expressing full-length engineered $FES^{S700C}$ is a valuable orthogonal method to standard biochemical assays using purified, truncated proteins.

Subsequently, it was investigated whether WEL028 could also engage $FES^{S700C}$ in living cells. To this end, HEK293T cells overexpressing $FES^{WT}$ or $FES^{S700C}$ were incubated with various concentrations of WEL028, after which cells were harvested and lysed. WEL028-labeled proteins were then visualized using click chemistry. Dose-dependent labeling of $FES^{S700C}$ was observed (Fig. 4f, g), which indicates that WEL028 is cell-permeable and can serve as a two-step probe in living cells. Of note, a concentration of 100 nM WEL028 was sufficient to fully block the labeling of $FES^{S700C}$. Autophosphorylation of Tyr713 on the activation loop of FES is a hallmark for its kinase activity[14,43]. Consequently, immunoblot analysis using a phospho-specific antibody for pY713 revealed that WEL028 (100 nM) fully abolished autophosphorylation of $FES^{S700C}$ but not $FES^{WT}$ (Fig. 4h, i). This indicates that the target engagement as measured by gel-based probe labeling assay correlates with the functional

**Table 1 Inhibitory potency of synthesized TAE684 derivatives against FES^WT and FES^S700C.**

| Compound | R$_1$ | R$_2$ | pIC$_{50}$ FES$^{WT}$ | pIC$_{50}$ FES$^{S700C}$ | Apparent fold selectivity |
|---|---|---|---|---|---|
| **1** (TAE684) | | | 8.1 ± 0.04 | 9.0 ± 0.02 | 8.6 |
| **2** | H | | 7.3 ± 0.06 | 7.8 ± 0.04 | 2.5 |
| **3** | H | | 6.5 ± 0.06 | 6.3 ± 0.06 | 0.65 |
| **4** | H | | 5.7 ± 0.21 | 8.4 ± 0.03 | 238 |
| **5** | H | | < 5 | 5.2 ± 0.08 | ND |
| **6** (WEL028) | | | 5.0 ± 0.37 | 8.4 ± 0.03 | 232 |

Half maximal inhibitory concentrations (expressed as pIC$_{50}$) determined on recombinantly expressed FES$^{WT}$ and FES$^{S700C}$ in a TR-FRET assay with 5 µM ATP. Apparent fold selectivity was calculated as IC$_{50}$ on FES$^{WT}$ divided by IC$_{50}$ on FES$^{S700C}$. Data represent means ± SD; $n = 3$. ND: not determined. Source data are provided as a Source Data file and complete dose–response curves can be found in Supplementary Fig. 1.

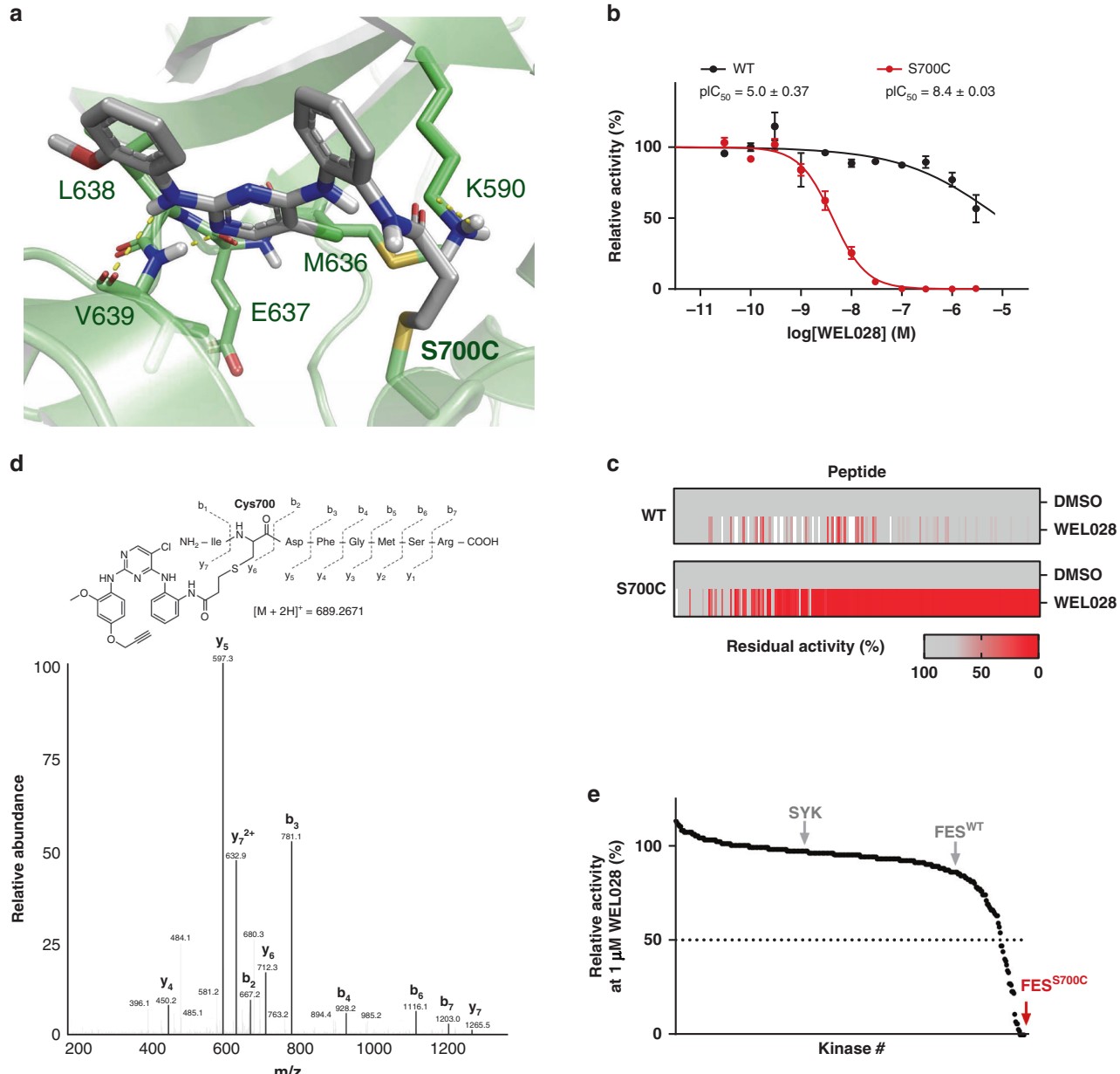

**Fig. 3 FES$^{S700C}$ is selectively and covalently inhibited by WEL028. a** Proposed covalent binding mode of compound 4 to Cys700 in crystal structure of FES (PDB code: 4e93). **b** In vitro inhibition profile of FES$^{WT}$ and FES$^{S700C}$ by WEL028 (TR-FRET assay, $n = 3$). **c** Inhibition profile of WEL028 (100 nM) on FES$^{WT}$ and FES$^{S700C}$ on peptide substrates (PamChip® microarray, normalized to vehicle-treated control, $n = 3$). Scale shows residual activity from minimum (0%, red) to maximum (100%, gray). **d** MS/MS-based identification of WEL028 covalently bound to Cys700. Precursor ion ($m/z$ [M + 2H]$^{2+}$ = 689.2671) was fragmented and signature ions are shown. Precursor ion was not observed in vehicle-treated control. **e** In vitro selectivity profile of WEL028 (1 μM, 1 h preincubation) on 380 recombinant kinases, visualized as waterfall plot (each data point is an individual kinase; $n = 2$). Data represent means ± SEM. Source data are provided as a Source Data file.

activity of FES$^{S700C}$ as determined in the biochemical and immunoblot assays.

**CRISPR/Cas9 gene editing for visualization of endogenous FES.** To obtain a physiologically relevant model system for studying target engagement and avoid transient overexpression, CRISPR/Cas9 gene editing was employed to introduce the S700C mutation endogenously in the human HL-60 promyeloblast cell line. Depending on the differentiation agent, HL-60 cells are capable to undergo differentiation along the monocyte/macrophage (16 nM PMA, 48 h) as well as neutrophil lineage (1 μM

ATRA, 1.25% DMSO, 72–96 h) (Fig. 5a)[44]. In addition, HL-60 cells have been widely used to study neutrophil function[45] as an experimentally tractable alternative for primary neutrophils, which have a short life-span and cannot be grown in cell culture[46].

To sensitize endogenously expressed FES in HL-60 cells to the mutant-specific probe, a single guide (sg)RNA target was selected with predicted site of cleavage in close proximity of the desired mutation in exon 16 of the *FES* locus (Fig. 5b). In conjunction, a single-stranded oligodeoxynucleotide (ssODN) homology-directed repair (HDR) donor was designed, aimed to introduce the target S700C mutation along with the implementation of a

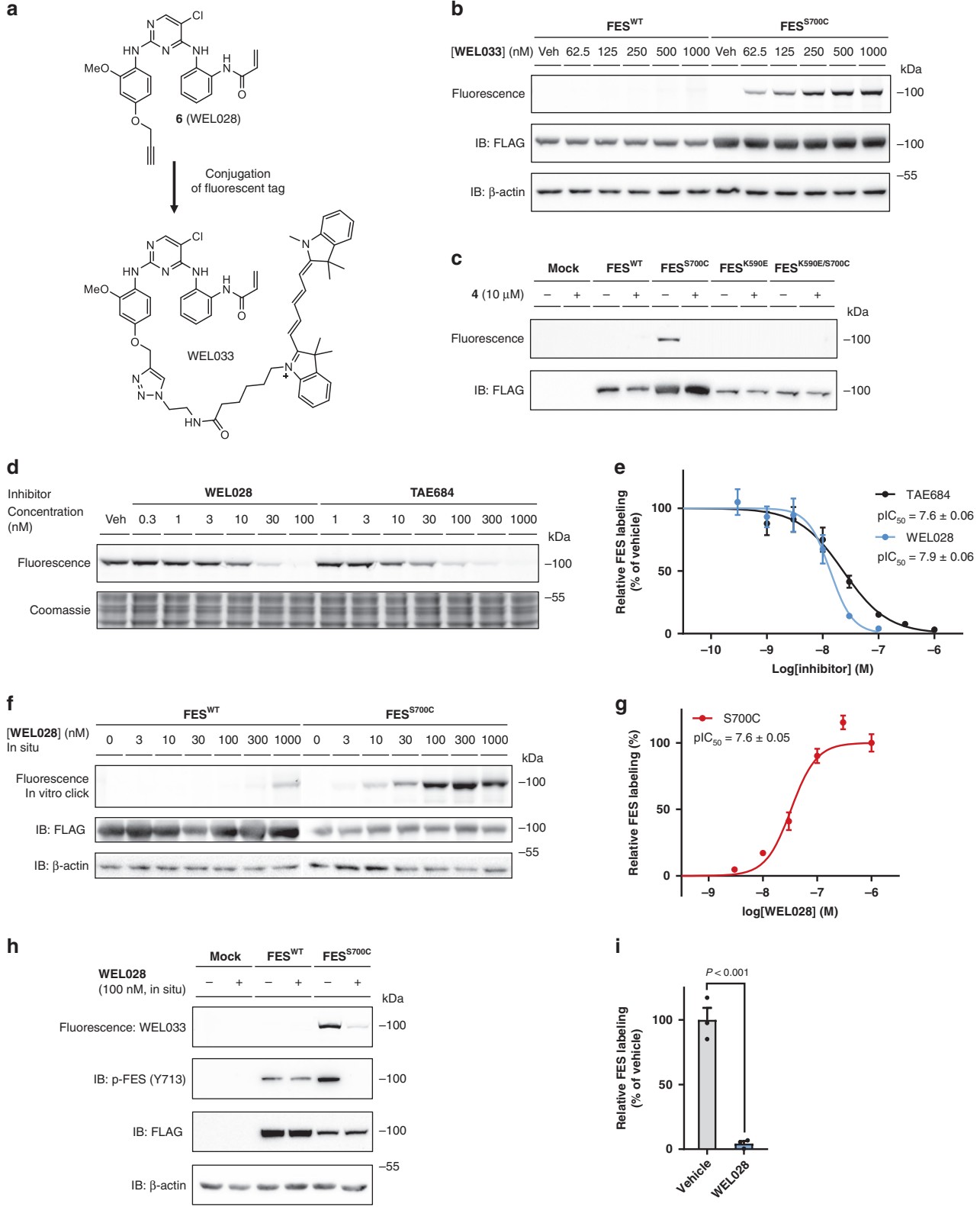

**Fig. 4 FES^S700C can be visualized by fluorescent probe WEL033. a** Design of fluorescent probe WEL033 (Cy5-conjugate). **b** Dose-dependent labeling of full-length FES^S700C by WEL033 in HEK293T cell lysate. **c** Labeling by WEL033 is specific, exclusive for FES^S700C and dependent on catalytic lysine 590. Lysates were preincubated with vehicle or 4 and labeled by WEL033 (250 nM). **d, e** Visualization of FES^S700C target engagement by WEL028 and TAE684. Lysate was preincubated with WEL028 or TAE684 and labeled by WEL033 (250 nM). Band intensities were normalized to vehicle-treated control (n = 3). **f, g** WEL028 engages recombinantly expressed FES^S700C in live cells. Transfected HEK293T cells were treated with WEL028 and labeled proteins were visualized using click chemistry. Band intensities were normalized to highest concentration (n = 3). **h, i** WEL028 blocks FES^S700C but not FES^WT autophosphorylation (visualized by immunoblot using anti-phospho-FES Y713). Band intensities were normalized to vehicle-treated control (n = 3). Data represent means ± SEM. Statistical analysis: two-tailed t-test: *** P < 0.001. Source data are provided as a Source Data file.

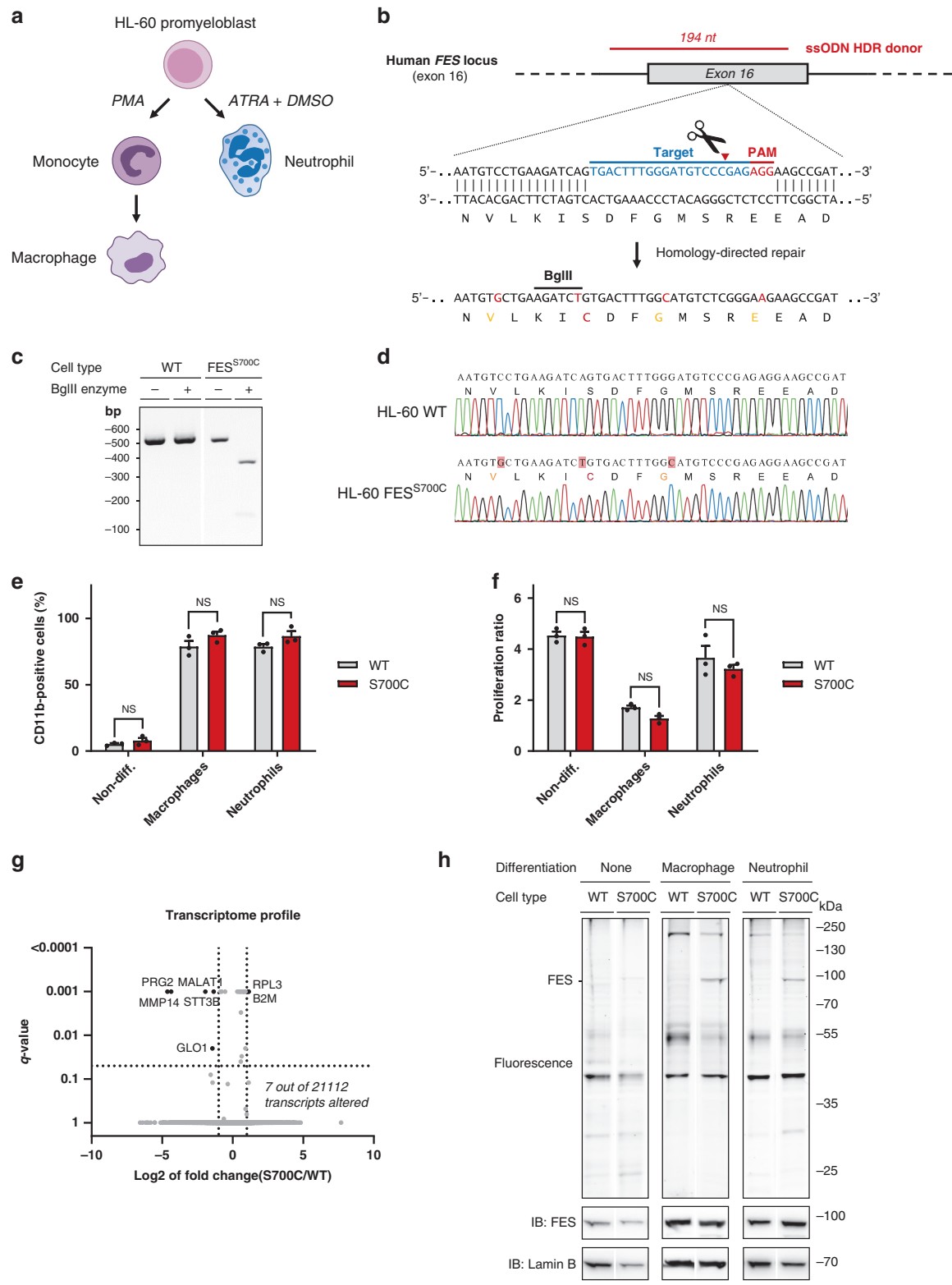

restriction enzyme recognition site to facilitate genotyping using a restriction fragment length polymorphism (RFLP) assay. Of note, the ssODN donor included also silent mutations to prevent cleavage of the ssODN itself or recleavage of the genomic locus after successful HDR (Fig. 5b). HL-60 cells were nucleofected with plasmid encoding sgRNA and Cas9 nuclease along with the ssODN donor, followed by single cell dilution to obtain clonal cultures. We identified one homozygous S700C mutant clone out

of approximately 100 screened clones by RFLP analysis (Fig. 5c). Sanger sequencing verified that the mutations had been successfully introduced without occurrence of undesired deletions or insertions (Fig. 5d). No off-target cleavage events were found in a predicted putative off-target site (Supplementary Table 5 and Supplementary Fig. 10).

Comprehensive biochemical profiling of FES^S700C showed no functional differences compared to FES^WT in any of the in vitro

**Fig. 5 Visualization of endogenous FES$^{S700C}$ in CRISPR/Cas9-edited HL-60 cells. a** HL-60 cells can differentiate into macrophages upon treatment with phorbol 12-myristate 13-acetate (PMA) for 48 h or into neutrophils with all-trans-retinoic acid (ATRA, 1 μM) and DMSO (1.25%) for 72–96 h. **b** CRISPR/Cas9 gene editing strategy. Selected sgRNA (bold blue) directs Cas9 to cleave at predicted site (red triangle). A ssODN homology-directed repair (HDR) donor template (red) flanks introduced mutations with 80 bp homology arms. The S700C mutation generates a BglII restriction site along with three silent mutations (orange) to remove PAM sites. Of note, two of the three mutated PAM sites correspond to sgRNAs not used in this study. PAM: protospacer-adjacent motif. **c** Restriction-fragment length polymorphism (RFLP) assay for identification of HL-60 FES$^{S700C}$ clone. Genomic region was amplified by PCR and amplicons were digested with BglII. Expected fragment size after digestion: 365 + 133 bp. **d** Sanger sequencing traces of WT HL-60 cells and homozygous FES$^{S700C}$ HL-60 clone. No deletions, insertions or undesired mutations were detected. **e** CD11b surface expression of HL-60 cells prior to and after differentiation, analyzed by flow cytometry. Threshold for CD11b-positive cells was determined using isotype control antibody ($n = 3$). **f** Reduction in proliferation upon differentiation is similar for WT and FES$^{S700C}$ cells. Proliferation ratio: cell number after differentiation divided by cell number before differentiation ($n = 3$). **g** Transcriptome profile of WT and FES$^{S700C}$ HL-60 macrophages analyzed by TempO-Seq. Differentially expressed genes (shown in black) were identified using following cut-offs: log2 of fold change FES$^{S700C}$ over WT HL-60 cells < −1 or > 1 with $q$-value < 0.05 ($n = 3$). **h** Endogenous FES is visualized by WEL033 in differentiated HL-60 FES$^{S700C}$ cells. WT or FES$^{S700C}$ HL-60 cells promyeloblasts, macrophages or neutrophils were lysed, followed by labeling with WEL033 (1 μM). FES expression increases upon differentiation (anti-FES immunoblot). Data represent means ± SEM. Statistical analysis: two-tailed t-test with Holm-Sidak's multiple comparisons correction: NS if $P > 0.05$. Source data are provided as a Source Data file.

assays (Fig. 2). In addition, we validated that HL-60 FES$^{S700C}$ cells differentiated into macrophages or neutrophils in an identical fashion as WT HL-60 cells. The percentage of differentiated cells after treatment with differentiation agents was quantified by monitoring surface expression of CD11b, a receptor present on HL-60 macrophages and neutrophils but not on non-differentiated HL-60 cells (Fig. 5e)[47]. No significant differences were observed between WT and FES$^{S700C}$ HL-60 cells. In line with this observation, WT and mutant cells undergoing differentiation demonstrated a similar decrease in proliferation (Fig. 5f). Upon differentiation along the macrophage lineage, HL-60 FES$^{S700C}$ cells acquired a typical monocyte/macrophage morphology (e.g. adherence to plastic surfaces, cell clumping and cellular elongation) comparable to WT cells (Supplementary Fig. 11). In a similar fashion, HL-60 FES$^{S700C}$ cells differentiated into neutrophils acquired the ability to induce a respiratory burst upon PMA stimulation, a characteristic phenotype of functional neutrophils (Supplementary Fig. 12,e, f). To confirm that the mutant HL-60 cell line exhibited minimal transcriptional alterations compared to parental WT cell line (e.g. due to clonal expansion), we performed a targeted transcriptomics analysis using the TempO-Seq technology (Fig. 5g)[48]. Only seven out of 21112 of the identified transcripts (0.03%) were significantly altered in FES$^{S700C}$ compared to WT HL-60 macrophages, which indicates that introduction of this mutation minimally disturbs gene expression. Notably, none of these genes are known to be involved in myeloid differentiation.

Next, cell lysates of macrophages or neutrophils derived from WT and FES$^{S700C}$ HL-60 cells were incubated with fluorescent probe WEL033 to visualize endogenous FES (Fig. 5h). In-gel fluorescence scanning of the WEL033-labeled proteome of FES$^{S700C}$ HL-60 neutrophils and macrophages revealed a band at the expected MW of FES (~93 kDa), which was absent in WT HL-60 cells. This fluorescent band was less prominent in non-differentiated HL-60 FES$^{S700C}$ cells, likely due to lower FES expression levels prior to differentiation (Fig. 5h, anti-FES immunoblot). Of note, WEL033 labeled a number of additional proteins (MW of ~200, ~55 and ~40 kDa, respectively) at the concentration used for FES detection (1 μM). In short, these results demonstrate that endogenously expressed engineered FES can be visualized using complementary chemical probes.

**Cellular target engagement in differentiating HL-60 cells.** FES was previously reported as an essential component of the cellular signaling pathways involved in myeloid differentiation[49,50]. However, most of these studies relied on the use of over-expression, constitutively active mutants, or antisense-based knockdown of FES. In addition, it remains unclear whether this

role of FES is dependent on its kinase activity. To revisit this question with pharmacological tools, it is key to establish which concentration of inhibitor is minimally required to fully inhibit the target in an endogenous setting. To this end, we tested whether WEL028 inhibited endogenously expressing FES$^{S700C}$ HL-60 cells at 100 nM and 1 μM during PMA-induced differentiation towards macrophages (Fig. 6a; Supplementary Fig. 13). Cells were harvested and lysed, followed by labeling of residual active FES$^{S700C}$ by WEL033, which revealed full target engagement of engineered FES at a concentration of 100 nM WEL028 (Fig. 6b), with only two prominent off-targets (~150 and ~40 kDa). At a higher concentration of 1 μM, WEL028 was substantially less selective (Fig. 6a) and inhibited the labeling of multiple proteins.

The percentage of differentiated cells after treatment with the differentiation agent was quantified by monitoring surface expression of CD11b[47]. Strikingly, despite complete target engagement of FES$^{S700C}$ at 100 nM WEL028 (Fig. 6b), the percentage of CD11b-positive cells was minimally affected (Fig. 6c-d). Cell proliferation, an indirect hallmark of differentiation, was decreased to identical levels for FES$^{S700C}$ HL-60 cells treated with vehicle or 100 nM WEL028 (Fig. 6e). Accordingly, FES$^{S700C}$ HL-60 cells treated with 100 nM WEL028 acquired macrophage morphology (e.g. adherence to plastic surfaces, cell clumping and cellular elongation) comparable to vehicle-treated controls (Supplementary Fig. 11). Together, these results show that complete FES inhibition does not affect PMA-induced differentiation of HL-60 cells into macrophages, suggesting that FES activity is dispensable for this process. In a similar fashion, FES activity was found to be dispensable for differentiation of HL-60 cells into functional neutrophils (Supplementary Fig. 12).

Remarkably, FES$^{S700C}$ cells undergoing PMA-induced differentiation in presence of a higher concentration of WEL028 (1 μM) completely failed to express CD11b, exhibited a less pronounced decrease in proliferation and displayed phenotypic characteristics similar to non-differentiated cells (Fig. 6c–e and Supplementary Fig. 11). Competitive probe labeling experiments revealed multiple WEL028 off-targets at 1 μM (Fig. 6a), which suggested that the observed block in differentiation might be due to off-target effects. A beneficial feature of the used chemical genetic strategy is that WT cells can account for these off-targets. Indeed, 1 μM WEL028 had similar effects on CD11b surface expression, proliferation and morphology of WT HL-60 cells subjected to differentiation (Fig. 6c–e and Supplementary Fig. 11). This verifies that the functional effects of WEL028 at 1 μM can indeed be attributed to off-target rather than on-target effects.

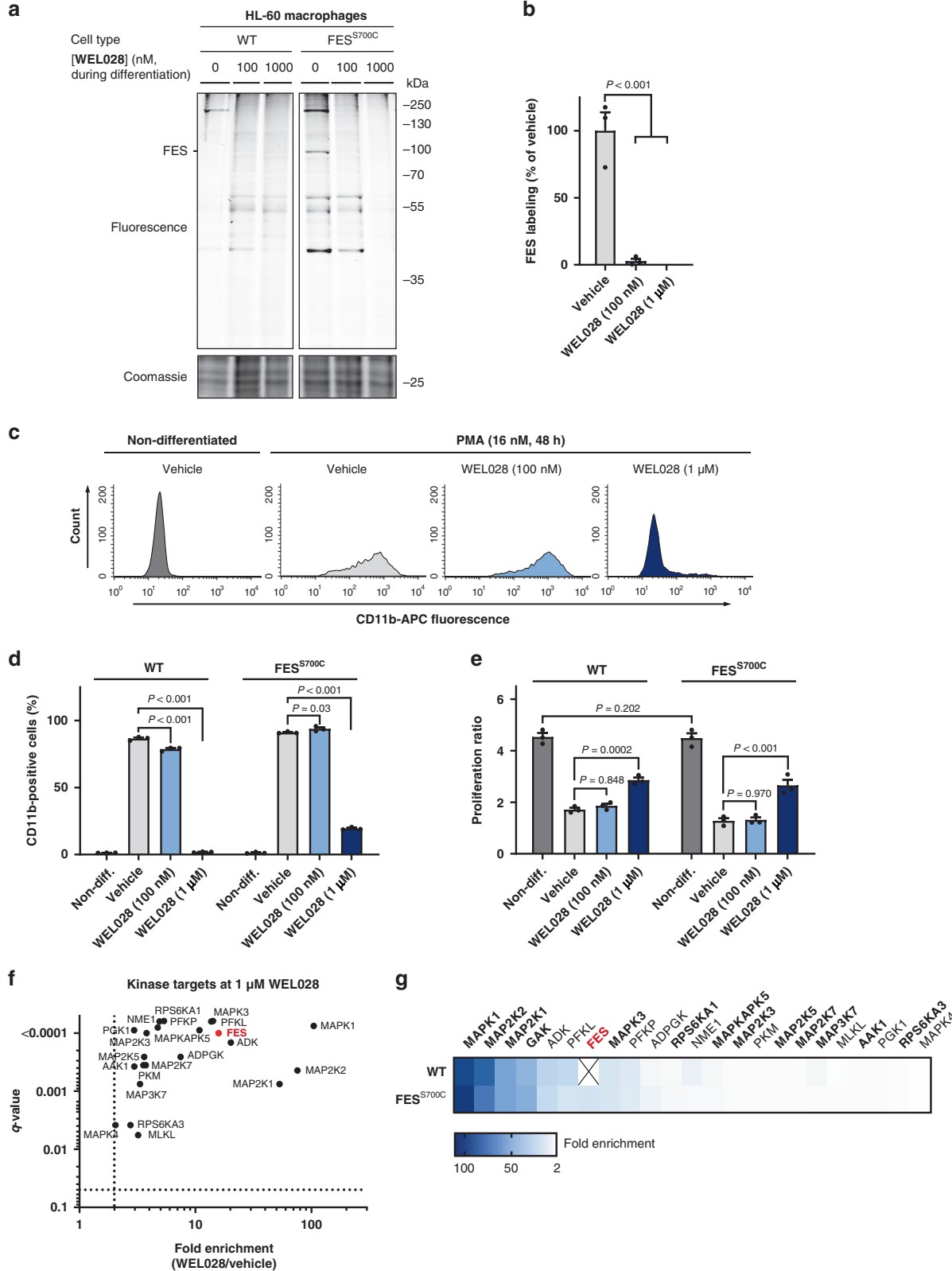

Quantitative label-free chemical proteomics was used to identify the off-targets of WEL028 at this high concentration. The WEL028-labeled proteome was conjugated to biotin-azide post-lysis, followed by streptavidin enrichment, on-bead protein digestion and peptide analysis using mass spectrometry. Significantly enriched kinases in WEL028-treated samples compared to vehicle-treated samples were designated as targets (Fig. 6f-g).

Identified off-targets included protein kinases harboring native cysteines at the DFG-1 position (depicted in bold), as well as several metabolic kinases (ADK, PFKL, PFKP, ADPGK, NME1, PKM, PGK1), although the latter lack a DFG motif. Notably, the off-target profile of WT and FES$^{S700C}$ HL-60 cells was identical with the exception of FES, which was exclusively present in mutant cells. Taken altogether, these results highlight that, despite

**Fig. 6 Comparative target engagement profiling in WT and FES$^{S700C}$ HL-60 cells during macrophage differentiation. a, b** Gel-based target engagement profile of WEL028 on WT and FES$^{S700C}$ HL-60 cells treated during PMA-induced differentiation. Cells were pretreated with vehicle or WEL028 (100 or 1000 nM, 1 h) prior to induction of differentiation towards monocytes/macrophages with PMA (16 nM, 48 h). Medium was refreshed with growth medium containing WEL028 and PMA after 24 h to maintain full FES inhibition. Lysates were post-labeled with WEL033 (1 μM, 30 min, rt). Band intensities were normalized to vehicle-treated control ($n = 3$). **c, d** CD11b surface expression analyzed by flow cytometry. Threshold for CD11b-positive cells was determined using isotype control antibody. Histograms illustrate representative replicates of FES$^{S700C}$ cells ($n = 3$). **e** Proliferation of WT and FES$^{S700C}$ HL-60 cells subjected to macrophage differentiation. Proliferation ratio: live cell number after differentiation divided by live cell number before differentiation ($n = 3$). **f** Chemical proteomics-based identification of WEL028 kinase targets at 1 μM in FES$^{S700C}$ HL-60 cells undergoing differentiation towards macrophs. Kinases with >2-fold enrichment compared to vehicle control ($q < 0.05$) were designated as targets. Values represent means of fold enrichment ($n = 3$). **g** Heat map representation of WEL028 (1 μM) target engagement profile in WT and FES$^{S700C}$ HL-60 cells undergoing macrophage differentiation, determined by chemical proteomics. Scale shows fold enrichment from minimum (2, white) to maximum (100, dark blue). Kinases with native cysteine at DFG-1 position are shown in bold. All data represent means ± SEM. Statistical analysis: two-tailed $t$-test with Holm-Sidak multiple comparisons correction: ***$P < 0.001$; *$P < 0.05$; NS if $P > 0.05$. Source data are provided as a Source Data file.

a limited number of off-targets, WEL028 can be effectively used in target engagement and validation studies using WT and FES$^{S700C}$ HL-60 cells.

**Role for FES in neutrophil phagocytosis via SYK activation.** Next, we sought to apply our chemical genetics strategy to study the role of FES in neutrophils, the first line of defense against invading bacteria. They are recruited to the site of infection and their primary function is to phagocytize and kill the pathogens. Neutrophil phagocytosis is a complex process that occurs via (a cross-talk of) various receptors, including pathogen-associated molecular pattern (PAMP) receptors, Fcγ receptors (FcγRs) and complement receptors (CRs)[51] Since FES was previously reported to regulate cell surface receptors, including TLR4 in macrophages[52] and FcεRI in mast cells[11,19], we wondered whether FES might be involved in neutrophil phagocytosis. First, we confirmed that treatment of live neutrophils with a low concentration of WEL028 (100 nM, 1 h) resulted in complete and selective inhibition of FES in the mutant cells (Fig. 7a-b). Partial inhibition of two off-targets (~80% inhibition of ~150 kDa protein and ~10% inhibition of ~40 kDa protein) in both WT and mutant cells was observed (Fig. 7a). To identify these off-targets, cells were harvested and the WEL028-bound targets were identified using chemical proteomics. FES was exclusively identified in FES$^{S700C}$ cells, whereas GAK and MAPK1 were identified as the only two enriched proteins in both WT and FES$^{S700C}$ neutrophils (Fig. 7c and Supplementary Fig. 14). Notably, the molecular weight of GAK (143 kDa) and MAPK1 (41 kDa) match the size of the two fluorescent bands visualized on gel, and both proteins were previously identified as WEL028 targets in the kinome screen (Supplementary Fig. 4). Taken together, these data indicated that our chemical genetics strategy using WEL028 can be used to study the role of FES in neutrophils.

To measure the phagocytic uptake by HL-60 neutrophils, a flow cytometry-based assay with live GFP-expressing *E. coli* was employed (Fig. 7d, e). Both WT and FES$^{S700C}$ neutrophils effectively internalized bacteria, with identical phagocytic indices. Control cells incubated on ice were included to account for surface binding without internalization. Interestingly, WEL028 at an adequate concentration (100 nM) for complete and selective FES inactivation (Fig. 7a), reduced the phagocytic index by 30–50% in FES$^{S700C}$ expressing cells, but not in WT HL-60 neutrophils (Fig. 7d, e, i; Supplementary Fig. 15). Since the off-targets GAK and MAPK1 are shared among WT and FES$^{S700C}$ HL-60 neutrophils (Fig. 7a and Supplementary Fig. 14), these results indicate that on-target FES inhibition is responsible for the observed reduction in phagocytosis of *E. coli*.

To gain insight in the molecular mechanisms of FES-mediated phagocytosis, we examined the previously obtained

substrate profile in more detail (Fig. 2e, Supplementary Table 1). A peptide of the non-receptor tyrosine kinase ZAP70 was identified as prominent FES substrate. Incubation with WEL028 abolished peptide phosphorylation by FES$^{S700C}$, but not FES$^{WT}$ (Fig. 7f). Although ZAP70 is predominantly linked to immune signaling in T-cells, its close homologue SYK is ubiquitously expressed in various immune cells, including neutrophils[46]. Moreover, SYK is part of signaling pathways linked to pathogen recognition and involved in bacterial uptake by neutrophils[53]. The identified ZAP70 peptide substrate shows high sequence similarity to its SYK counterpart surrounding Y352 (Fig. 7f). To validate that SYK is a downstream target of FES, SYK-V5 and FES$^{S700C}$-FLAG were co-transfected in U2OS cells. First, it was confirmed that overexpression of FES$^{S700C}$ led to autophosphorylation of FES at Y713, which was sensitive to WEL028 (Fig. 7g). Subsequent immunoblot analysis using a SYK Y352 phospho-specific antibody showed that SYK was phosphorylated in a FES-dependent manner (Fig. 7g). In accordance, co-transfection of SYK with a kinase-dead FES$^{K590E}$ variant abolished Y352 phosphorylation (Supplementary Fig. 16a). Of note, WEL028 did not inhibit SYK in the kinome screen (Supplementary Fig. 4) and did not affect SYK pY352 levels upon co-transfection with FES$^{WT}$ (Supplementary Fig. 16a) or in absence of FES (Supplementary Fig. 16b). In addition, immunoprecipitation against FES$^{S700C}$ using an anti-FLAG antibody revealed a physical interaction between FES and SYK as witnessed by immunoblot against the V5-tag of SYK (Fig. 7h). Importantly, this interaction was dependent on the activation status of FES, because WEL028 inhibited the co-precipitation of SYK with FES. Taken together, these results suggest that SYK Y352 is a direct substrate of FES.

To verify that endogenous SYK is also involved in phagocytic uptake of *E. coli* by neutrophils, HL-60 neutrophils were incubated with the potent SYK inhibitor R406. SYK inhibition reduced phagocytosis to similar levels as observed for WEL028 (Fig. 7i). Of note, U-73122 and Cytochalasin D were taken along as positive controls to verify that the phagocytosis is mediated via phospholipase C gamma 2 (PLCγ2) and actin polymerization, respectively[51]. Finally, we verified whether the downstream signaling pathway of SYK was modulated in a FES-dependent manner by using immunoblot analysis with phospho-specific antibodies for SYK Y352, hematopoietic cell-specific protein-1 (HS1) Y397 (an actin-binding protein) and PLCγ2 Y1217. Indeed, a transient phosphorylation of SYK, HS1, and PLCγ2 was observed, which peaked at 2 min after incubation with bacteria (Supplementary Fig. 17). Treatment with WEL028 completely blocked the phosphorylation of these proteins (Fig. 7j) in FES$^{S700C}$ but not WT neutrophils, confirming an on-target, FES-specific effect. As an

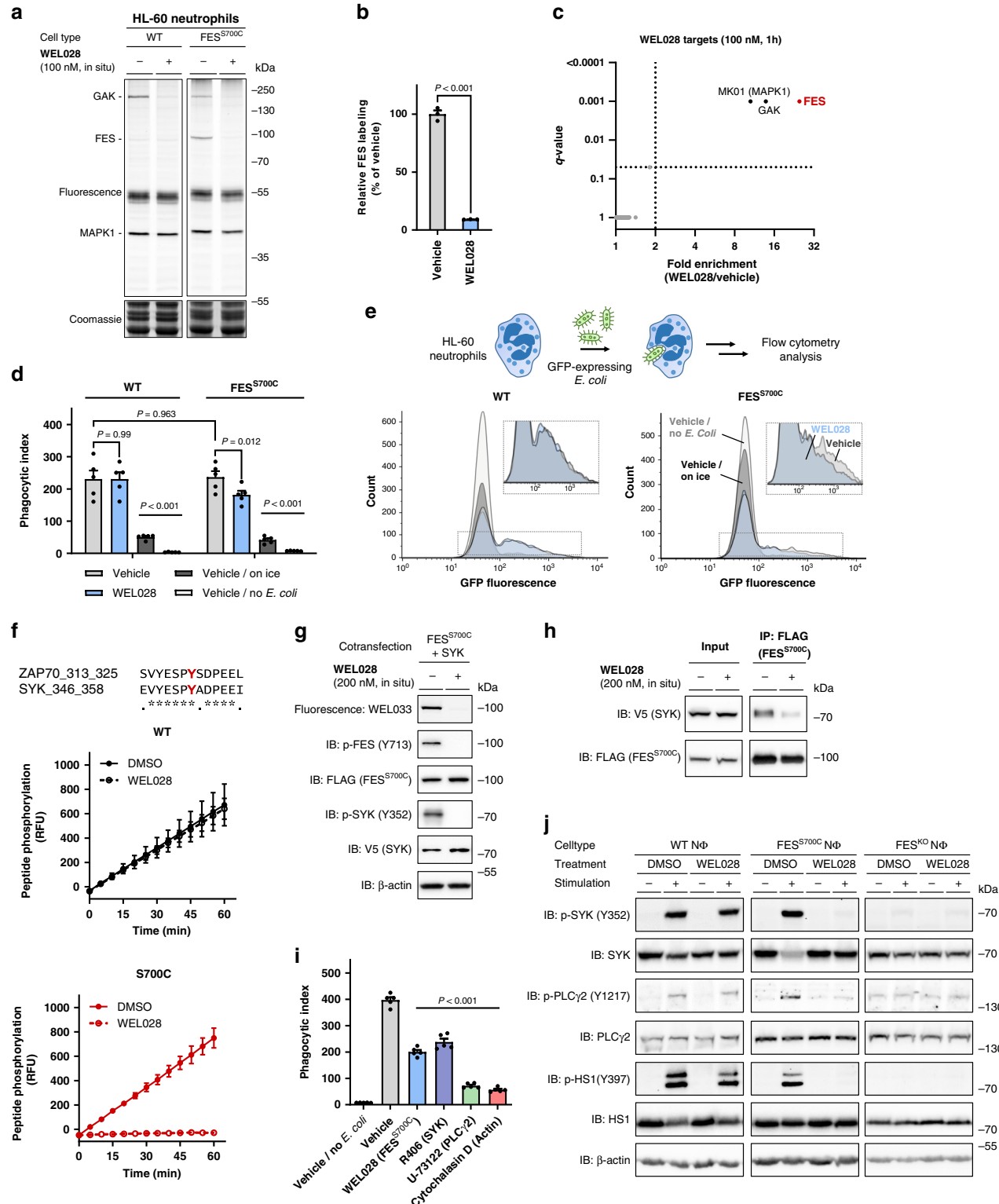

independent, alternative genetic method to validate the role of FES in this signaling pathway, we generated a FES knockout HL-60 cell line (Supplementary Fig. 18, a–d and Supplementary Table 6). Notably, these FES^KO cells maintained the ability to differentiate into neutrophils (Supplementary Fig. 18e), but showed severely impaired SYK, HS1, and PLCγ2 phosphorylation. Taken together, these results indicate that FES activity is required for the activation of this signaling pathway in HL-60 neutrophils in response to *E. coli* infection.

## Discussion

The field of chemical genetics has previously generated tools to aid in kinase target validation[54,55]. An example is the powerful "analog-sensitive" (AS) technology, where the gatekeeper residue is changed into a less bulky residue, enabling the kinase of interest to accommodate bulky ATP analogs in its active site[56]. However, these analogs do not form covalent adducts with the kinase and therefore do not allow visualization of target engagement. Furthermore, mutagenesis of gatekeeper residues may result in

**Fig. 7 FES mediates _E. coli_ phagocytosis by HL-60 neutrophils via SYK activation. a, b** Complete FES$^{S700C}$ inhibition at 100 nM WEL028 in HL-60 neutrophils. Cells were differentiated into neutrophils (1 μM ATRA, 1.25% DMSO, 72–96 h), treated with WEL028 (100 nM, 1 h) and lysates were post-labeled (1 μM WEL033). Band intensities normalized to vehicle-treated control ($n = 3$). **c** Identification of WEL028 (100 nM) kinase targets in FES$^{S700C}$ HL-60 neutrophils using chemical proteomics. Kinases with >2-fold enrichment compared to vehicle control ($q < 0.05$) were designated as targets (means of fold enrichment, $n = 4$). **d–e** WEL028 reduces phagocytic uptake in FES$^{S700C}$ but not WT neutrophils. Treatment as in (**a**)–(**b**), followed by addition of GFP-expressing _E. coli_ B834 (MOI = 30, 1 h, 37 °C) and flow cytometry analysis ($n = 5$). Phagocytic index: fraction GFP-positive cells (number of phagocytic cells) multiplied by GFP MFI (number of phagocytized bacteria). **f** SYK Y352 as proposed FES phosphorylation site (bold red), based on phosphorylation of homologous ZAP70 peptide (PamChip® microarray, $n = 3$). **g** FES phosphorylates SYK Y352 in situ. U2OS cells co-expressing FES$^{S700C}$-FLAG and SYK-V5 were incubated with vehicle or WEL028 (200 nM, 1 h). Lysates were labeled (250 nM WEL033) and analyzed by in-gel fluorescence and immunoblot ($n = 3$). **h** SYK interacts with FES. Co-transfected U2OS cells were incubated as in (**g**), followed by anti-FLAG immunoprecipitation and immunoblot analysis ($n = 3$). **i** Phagocytosis of _E. coli_ by HL-60 neutrophils depends on FES, SYK and PLCγ2 activity and actin polymerization. Incubations as in (**d**)–(**e**) (WEL028: 100 nM, R406: 1 μM, U-73122: 5 μM, cytochalasin D: 10 μM; $n = 5$). **j** Phosphorylation of SYK Y352 and downstream substrates is inhibited by WEL028 in infected FES$^{S700C}$ but not WT neutrophils, and absent in FES$^{KO}$ neutrophils. Inhibitor incubations as in (**d**)-(**e**), followed by addition of GFP-expressing _E. coli_ B834 (MOI = 30, 2 min, 37 °C) and immunoblot analysis ($n = 3$). Data represent means ± SEM. Statistical analysis: ANOVA with Holm-Sidak's multiple comparisons correction: ***$P < 0.001$; *$P < 0.05$; NS if $P > 0.05$. Source data are provided as Source Data file.

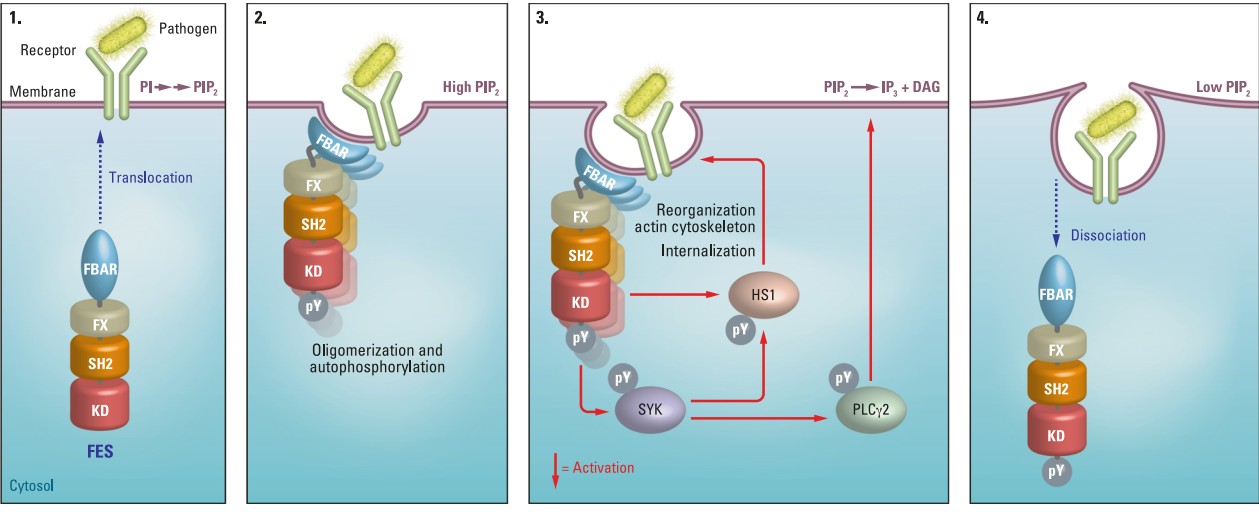

**Fig. 8 Proposed model for the role of FES in neutrophil phagocytosis via SYK-HS1-PLCγ2 pathway activation.** Representations are simplified for illustration purposes.

impaired catalytic activity. The suboptimal pharmacokinetic properties of ATP analogs used in the AS technology limit their applicability for in vivo target validation studies[57]. The concept of "covalent complementarity" is based on mutagenesis of the gatekeeper[35] or gatekeeper+6[37] residue into a cysteine to function as nucleophile. Although this allows the development of covalent probes for target engagement studies, a secondary mutation in the active site was required to improve gatekeeper cysteine reactivity or compound selectivity and potency[35,36]. This is particularly challenging when moving to an endogenous model system, since it would involve two independent CRISPR/Cas9 gene editing events to introduce these two mutations.

Here, we identified the DFG-1 residue as an excellent position for introducing a nucleophilic cysteine to react with an acrylamide as a complementary warhead, with no need for secondary point mutations to improve cysteine reactivity or inhibitor selectivity. Furthermore, mutagenesis of the DFG-1 position into a cysteine is functionally silent: it does not affect FES catalytic activity nor its substrate recognition and SH2 domain binding profile. FES$^{S700C}$ showed a minor increase in ATP-binding affinity, but this difference in $K_M$ is unlikely to have any consequences at physiologically relevant ATP concentrations, which are typically in the millimolar range[40]. Although nearly 10% of all known kinases have a native DFG-1 cysteine residue, many kinases harboring a DFG-1 cysteine showed limited or no inhibition by WEL028 (Supplementary Fig. 4 and Fig. 7c), suggesting that the choice of the chemical scaffold constitutes an

additional selectivity filter. An acrylamide group was selected as the electrophile to react with the intended cysteine, since it exhibits sufficient reactivity towards cysteines only when appropriately positioned for a Michael addition reaction and has limited reactivity to other intracellular nucleophiles[58]. This may additionally prevent non-specific interactions with targets outside of the kinase cysteinome. Given the selectivity profile and cellular permeability of WEL028, we postulate that the diaminopyrimidine scaffold is a useful addition to the toolbox of covalent complementary probes applied in chemical genetic strategies, which previously consisted of mainly quinazolines and pyrazolopyrimidines[35,37].

Previously reported chemical genetic methods relied on overexpression systems that disturb signal transduction cascades[39]. In contrast, a major benefit of our strategy lies in the control that our method allows to exert over a biological system without disturbing the cellular homeostasis. By changing a single base-pair in the _FES_ gene, we substituted a single atom (oxygen to sulfur) in the endogenous protein. Yet, this allowed us to rationally design and synthesize a chemical probe that visualizes and inhibits the engineered kinase activity in human cells. Arguably, this minimal change at the genome and protein level ensures that regulation at transcriptional and (post)-translational level activity are minimally disturbed. In fact, we could demonstrate that the mutant protein activity, substrate preference and protein–protein interactions were similar to the WT protein. Furthermore, WT and mutant cells behaved similarly in various functional assays

(e.g. proliferation, differentiation, and phagocytosis). Targeted sequencing experiments further confirmed minimal transcriptional differences (0.03% of read transcripts) between the clonal FES$^{S700C}$ cell line and parental WT cell line. Although the low efficiency of HDR-mediated mutagenesis did not allow us to identify more than one homozygous mutant clone, recent developments in CRISPR/Cas9 gene editing, base editing[59] and prime editing[60] will undoubtedly improve this efficiency in the future.

Our chemical genetics strategy allows temporal control in modulating kinase activity, opposed to conventional genetic approaches. In this report, we leverage this advantage by inactivating FES activity both during myeloid differentiation and in terminally differentiated neutrophils. In contrast to previous studies relying on overexpression of (constitutively active) FES or genetic knockout, we found that FES activity is dispensable for differentiation towards macrophages in HL-60 cells. These results are in line with two previous studies using genetic mouse models[17,61]. Caution should thus be taken when using overexpression systems, especially in case of artificial mutant kinases with constitutive activity, as these may induce artefacts in cellular physiology. It remains to be investigated whether FES plays a role in myeloid differentiation into other cell types, or independently of its kinase activity, e.g. as an interaction partner for other proteins.

We illustrated that comparative target engagement profiling in mutant and WT cells is a powerful approach to distinguish on-target from off-target effects. Our results highlight the relevance of visualizing target engagement to select a dose that is sufficient to completely inactivate the kinase of interest, and avoid doses that induce off-target effects. For example, differentiation of HL-60 cells was prevented at a higher concentration of WEL028 than required for complete FES inhibition. Chemical proteomics identified various members of the MAP kinase family as off-targets under these conditions. Interestingly, MAPK1/3 and MAP2K1/2 are reported to be essential for HL-60 cell differentiation along the monocyte/macrophage lineage[47].

Our chemical genetics strategy revealed a biological role for FES in neutrophils. Our data suggest that FES plays a role in the phagocytic uptake of bacteria in neutrophils by activating SYK and downstream substrates HS1 and PLCγ2. These data are in line with previous studies that reported a role for FES in regulating surface expression of adhesion molecule PSGL-1 in leukocytes[21] and TLR4 during bacterial phagocytosis by peritoneal macrophages[52] In combination with data reported in the literature (reviewed in ref. [11]), we can propose the following model (Fig. 8 and Supplementary Discussion). One of the first events in response to bacterial recognition by surface receptors is the formation of phosphatidylinositol 4,5-bisphosphate (PIP$_2$) in the membrane (Fig. 8a)[62]. FES normally resides in the cytosol in an inactive conformation, but translocation to the PIP$_2$-rich membrane may occur by binding via its F-BAR domain[19]. This likely triggers the formation of oligomers and auto-activation by phosphorylation on FES Y713[10] and induces membrane curvature required for particle internalization (Fig. 8b)[63]. FES may subsequently activate SYK by phosphorylation of Y352 (or perhaps indirectly via another kinase), which poses an alternative activation mechanism of SYK compared to the traditional activation via binding to ITAM domains of immunoreceptors[64,65]. SYK is known to phosphorylate HS1, an actin-binding protein involved in reorganization of the actin cytoskeleton. It can be phosphorylated on multiple tyrosine residues that all contribute to its actin remodeling function[66]. Of note, its Y378 and Y397 residues are phosphorylated by FES in mast cells, but both sites have also been identified as substrate sites for other kinases, including SYK[19]. Phosphorylation of HS1 by FES and/or SYK

drives reorganization of the actin cytoskeleton required for internalization of the bacterium-receptor complex (Fig. 8c). Concomitantly, the phosphorylated Y352 residue in SYK could serve as binding site for the SH2 domain of PLCγ2, followed by SYK-mediated PLCγ2 activation[67]. In turn, this would allow for degradation of PIP$_2$ into diacylglycerol (DAG) and inositol-triphosphate (IP$_3$), altering the membrane composition[62] and returning it to the non-activated state: FES dissociates from the membrane and the signaling process is terminated (Fig. 8d). This model thus proposes a feedback mechanism in which FES indirectly regulates its own localization and activation by modulating PLCγ2 activity via SYK.

Although we demonstrate that our strategy employing DFG-1 cysteine mutants can be applied to kinases other than FES (Supplementary Note 1), it is unlikely that complete kinome coverage can be achieved with a single chemotype of complementary probes. Structural information of the target of interest will facilitate the design of other complementary probe-protein pairs. It should be noted that application of this chemical genetic toolbox on other kinases should be accompanied by biochemical characterization of the corresponding mutant kinase to verify that its function is not affected. Consequently, it is important to confirm that CRISPR/Cas9-generated mutant cells behave similarly to WT cells in any employed downstream functional assays.

Finally, the selectivity acquired by combining gene editing and a complementary probe brings the advantages of acute, pharmacological inhibition without the need for extensive hit optimization programs to identify compounds of adequate potency and selectivity. Although we provided a proof of concept using a cell line as model system, tremendous advancements in gene editing technologies also provide means to generate mutant (stem) cells or animal models. In conclusion, we envision that the presented methodology could provide powerful pharmacological tools to study the target engagement and function of poorly characterized kinases and aid in their validation as therapeutic targets.

## Methods

**Materials**. All chemicals were purchased at Sigma Aldrich, unless stated otherwise. DNA oligos were purchased at Sigma Aldrich or Integrated DNA Technologies and sequences can be found in Supplementary Table 7. Cloning reagents were from Thermo Fisher. TAE684 and R406 were purchased at Selleckchem and Cytochalasin D and U-73122 at Focus Biomolecules. Cy5-azide and BODIPY-azide were previously synthesized in-house and characterized by NMR and LC-MS. All cell culture disposables were from Sarstedt. Bacterial and eukaryotic protease inhibitor cocktails were obtained from Amresco.

**Cloning**. Full-length human cDNA encoding FES and PAK4 was obtained from Source Bioscience. pDONR223-constructs with full-length human cDNA of FER, LYN, PTK2 and SYK were a gift from William Hahn & David Root (Addgene Human Kinase ORF Collection). For bacterial expression constructs, human FES cDNA encoding residues 448–822 or human FER cDNA encoding residues 448–820 was amplified by PCR and cloned into expression vector pET1a in frame of an N-terminal His$_6$-tag and Tobacco Etch Virus (TEV) recognition site. Eukaryotic expression constructs of FES, FER and PAK4 were generated by PCR amplification and restriction/ligation cloning into a pcDNA3.1 vector, in frame of a C-terminal FLAG-tag. Eukaryotic expression constructs of LYN, PTK2 and SYK were generated using Gateway™ recombinational cloning into a pcDest40 vector, in frame of a C-terminal V5-tag, according to recommended procedures (Thermo Fisher). Point mutations were introduced by site-directed mutagenesis and all plasmids were isolated from transformed XL-10 competent cells (prepared using *E. coli* transformation buffer set; Zymo Research) using plasmid isolation kits following the supplier's protocol (Qiagen). All sequences were verified by Sanger sequencing (Macrogen).

For CRISPR/Cas9 plasmids, guides were cloned into the BbsI restriction site of plasmid px330-U6-Chimeric_BB-CBh-hSpCas9 (gift from Feng Zhang, Addgene plasmid #42230).

**Protein expression and purification**. Bacterial expression constructs were transformed into *E. coli* BL21(DE3) co-transformed with a pCDFDuet-1 vector encoding *Yersinia* phosphatase YopH (kindly provided by prof. dr. Kuriyan). Cells

were grown in Luria Broth (LB) medium containing 50 µg/mL kanamycin and 50 µg/mL streptomycin at 37 °C to an $OD_{600}$ of 0.4. The cultures were cooled on ice and protein expression was induced by addition of 50 µM isopropyl-β-D-thiogalactopyranoside (IPTG) for 16 h at 18 °C. Cultures (typically 10 mL per mutant) were centrifuged (1000g, 10 min, 4 °C) and washed in 1 mL physiological salt solution (0.9% (w/v) NaCl). The pellet was then resuspended in 400 µL lysis buffer (100 mM $NaH_2PO_4$ pH 8.0, 500 mM NaCl, 10% glycerol, 0.5 mM tris(2-carboxyethyl)phosphine (TCEP), 20 mM imidazole, 1 x bacterial protease inhibitor cocktail) and cells were lysed by sonication on ice (15 cycles of 4″ on, 9.9″ off at 25% maximum amplitude; Vibra Cell (Sonics)). $MgCl_2$ and OmniCleave (Epicentre) were added to final concentrations of 10 mM and 125 U/mL, respectively, and lysates were incubated for 10 min at rt. Meanwhile, 200 µL Nickel Magnetic Beads (BiMake), were homogenized by vortexing and washed in lysis buffer (3 × 200 µL) using a magnetic separator. Lysate was added to beads and incubated for 1 h at 4 °C with vigorous shaking. Beads were then washed with lysis buffer containing 75 mM imidazole (3 × 400 µL), after which the beads were transferred to a clean Eppendorf tube. Protein was phosphorylated on beads by addition of autophosphorylation buffer (50 mM HEPES pH 7.5, 500 mM NaCl, 5% glycerol, 10 mM dithiothreitol (DTT), 10 mM $Na_3VO_4$, 2 mM $(NH_4)_2SO_4$, 10 mM $MgCl_2$, 5 mM $MnCl_2$, 2 mM ATP) and vigorous shaking for 2 h at rt. Beads were subsequently washed with lysis buffer containing 20 mM imidazole (3 × 400 µL), after which the protein was eluted in lysis buffer containing 250 mM imidazole (2 × 200 µL). The elution fractions were combined, applied onto a 10 kDa cutoff centrifugal filter unit (Amicon) and centrifuged (14,000g, 10 min, 4 °C). The retentate was reconstituted in 100 µL protein storage buffer (10 mM HEPES pH 7.5, 500 mM NaCl, 5% glycerol, 10 mM DTT), aliquoted and stored at −80 °C. Protein concentration was measured using Qubit fluorometric quantitation (Thermo Fisher) and protein purity was monitored by SDS-PAGE and Coomassie staining.

**In vitro TR-FRET kinase assay**. Assays were performed in white ProxiPlate-384 Plus™ 384-well microplates (Perkin Elmer). Incubation steps were performed at 21 °C in kinase reaction buffer (50 mM HEPES pH 7.5, 150 mM NaCl, 10 mM $MgCl_2$, 1 mM EGTA, 2 mM DTT, 0.01% Tween-20). Purified proteins were diluted in kinase reaction buffer prior to use. All measurements were performed in triplicate.

Plates were read on a TECAN Infinite M1000 Pro plate reader, using fluorescence top reading settings ($\lambda_{ex} = 320/20$ nm, $\lambda_{em,donor} = 615/10$ nm, $\lambda_{em,acceptor} = 665/10$ nm, 100 µs integration time, 50 µs lag time). Emission ratios were calculated as fluorescence of acceptor / fluorescence of donor. Z′-factors were determined for each individual assay plate as $Z' = 1 - 3(\sigma_{pc} + \sigma_{nc})/(\mu_{pc} - \mu_{nc})$, with σ = standard deviation and µ = mean of measured replicates, WT or untreated samples as positive control (pc) and samples incubated with no ATP as negative control (nc). All plates met the requirement of Z′ > 0.7. Emission ratios were corrected for background signal of samples incubated with no ATP. Corrected ratios were averaged and normalized to WT or untreated signal as 100% reference.

For relative activity determination of mutants, purified WT or mutant FES (12.5 ng per well) or FER (5 ng per well) was incubated with 50 nM ULight-TK peptide (Perkin Elmer) and 100 µM ATP for 60 min at 21 °C in a total volume of 10 µL. The reaction was quenched by addition of 10 µL development solution (20 mM EDTA, 4 nM Europium-anti-phosphotyrosine antibody (PT66, Perkin Elmer) and incubated for 60 min before fluorescence was measured.

For $K_M$ determinations, assay was performed as described above, but with variable ATP concentrations in a dilution series of 1 mM to 100 nM. For $IC_{50}$ determinations, serial dilutions of inhibitor were prepared in DMSO, followed by further dilution in kinase reaction buffer. The inhibitors were premixed with peptide and ATP (5 µM final concentration, unless indicated otherwise), after which WT or mutant kinase was added to initiate the reaction. The final DMSO concentration during the reaction was 1%. $K_M$ and $IC_{50}$ curves were fitted using GraphPad Prism® 7/8 (GraphPad Software Inc.).

**PamChip® microarray assays**. Kinase activity profiles were determined using the PamChip® 12 protein tyrosine (PTK) peptide microarray system (PamGene International B.V., 's-Hertogenbosch, The Netherlands) according to the instructions of the manufacturer. Arrays were blocked with 2% BSA (30 µL, 15 min, 2 cycles/min) and washed three times with kinase assay buffer (50 mM Tris pH 7.5, 10 mM $MgCl_2$, 1 mM EDTA, 2 mM DTT, 0.01% Brij-35, 7.5 µg/mL PY20-FITC antibody (AbD Serotec/Bio-Rad). Sample input was 0.25 ng purified FES (WT or S700C) per array and [ATP] = 400 µM. For arrays with inhibitor, recombinant FES was preincubated in assay buffer without ATP with vehicle or WEL028 (100 nM, 30 min, on ice, 2% final DMSO concentration). Peptide phosphorylation was monitored during the incubation with assay mixture, by taking images every 5 min at 50 ms exposure time, allowing real time recording of the reaction kinetics (one-step reaction). After washing of the array, fluorescence was detected at different exposure times (20, 50, 100, and 200 ms).

For SH2 domain binding assays, PamChip® protein tyrosine kinase (PTK) arrays were blocked by pumping a 2% BSA solution up and down through the array (30 µL, 15 min, 2 cycles/min). After three washing steps, the arrays were pre-phosphorylated by incubation with 1.5 pmol of (His₆-tagged) JAK2 catalytic domain for 60 cycles in kinase assay buffer with 0.01% BSA. After three washing steps, the arrays were incubated with 0.75 ng of FES (WT or S700C) in PBS/0.1% Tween supplemented with 0.01% BSA. Incubations without FES served as negative

control. After three washing steps, binding of FES was visualized by incubation with Alexa Fluor 488 labeled anti-penta-His antibody (7.5 µg/mL final concentration, Qiagen, #1019199) in PBS/0.01% Tween. Peptide phosphorylation of JAK2 was detected with the same anti p-Tyr antibody as used in the PTK activity assay. After incubation for 30 min (2 cycles/min), arrays were washed, and images taken at different exposure times.

Data quantification of the images at all exposure times and reaction times and visualization of the data were performed using the BioNavigator 6.3 software (PamGene International B.V., 's-Hertogenbosch, The Netherlands). Post-wash signals (local background subtracted) were used. After signal quantification and integration of exposure times, signals were log2-transformed for visualization. Peptides with no ATP-dependent signal were excluded from analysis. Identification of peptides that were significantly different between conditions was performed using a Mixed Model statistical analysis. Substrate consensus motif was generated using Enologos (http://www.benoslab.pitt.edu).

**Selectivity profiling**. Selectivity profiling assays were performed by the Invitrogen SelectScreen™ Services. A complete list of tested kinases can be found in Supplementary Fig. 4, detailed assay procedures are described in SelectScreen Assay Conditions documents located at www.invitrogen.com/kinaseprofiling. All kinases were preincubated for 1 h at indicated concentrations of inhibitor. The concentration of ATP was selected to be equal to the $K_M$, unless indicated otherwise. An initial screen was performed on 279 kinases at single dose of 1 µM. All available kinases with native cysteine residues in the active site were included in this panel. Kinases showing >50% inhibition at 1 µM were further profiled in dose-response experiments in a 10-point dilution series of 10 µM to 0.5 nM.

**Docking studies**. All structure-based modeling was performed in Schrödinger Suite (version 2017-2, Schrödinger). The docking of compound **4** was based on the crystal structure of FES co-crystallized with TAE684 (PDB: 4e93)[28], which was prepared by the protein preparation wizard. Prior to docking, the Ser700 residue was manually changed into a cysteine. Subsequently, compound **4** was aligned to TAE684 on the basis of the diaminopyrimidine (both ligands share the same kinase/hinge binding moiety). This pose was optimized using an exhaustive hierarchical optimization procedure available in Prime (version 2017-2, Schrödinger)[68] The acrylamide warhead was found in proximity of Cys700 and the ligand was then covalently attached to this residue, followed by another round of hierarchical optimization[68]. Figures were rendered using PyMOL Molecular Graphics System (version 1.8, Schrödinger).

**MS identification of covalent inhibitor-peptide adduct**. Purified FES^S700C (1.2 µg in 38 µL) was treated with WEL028 (2 µL of 20× concentrated stock in DMSO, 1 µM final concentration) or vehicle (as negative control) for 30 min at rt. The reaction was quenched by addition of 3× Laemmli buffer (20 µL), incubated for 5 min at 95 °C and sample (400 ng, 20 µL) was resolved on 10% acrylamide SDS-PAGE gel (200 V, 60 min). Gel was stained using Coomassie Brilliant Blue R-250, bands were cut out and gel into small blocks and destained in 500 µL of 50% MeOH in 100 mM $NH_4HCO_3$ pH 8.0 for 10 min at rt. Acetonitrile (500 µL) was added for gel blocks dehydration, after which gel blocks were digested with sequencing-grade trypsin (Promega) in 250 µL trypsin buffer (100 mM Tris pH 7.5, 100 mM NaCl, 1 mM $CaCl_2$, 10% acetonitrile) overnight at 37 °C with vigorous shaking. The pH was adjusted to pH 3, after which the sample was diluted in extraction solution (65% acetonitrile/35% MilliQ) and concentrated in a SpeedVac concentrator. Samples were subsequently desalted using StageTips with $C_{18}$ material. Each StageTip was conditioned with MeOH (50 µl, 300g, 2 min), followed by solution B (50 µl, 80% v/v acetonitrile, 0.5% v/v formic acid in MilliQ, 300g, 2 min) and solution A (50 µl, 0.5% v/v formic acid in MilliQ, 300g, 2 min). Next, samples were loaded, centrifuged (300g, 2 min) and the peptides on the StageTip were washed with solution A (100 µl, 300g, 2 min). The StageTips were transferred to new 1.5 mL low-binding tubes and peptides were eluted with solution B (100 µl, 300g, 2 min). Solvents were evaporated to dryness in a SpeedVac concentrator (45 °C, 3 h). Samples were reconstituted in LC-MS solution (50 µl, 3% v/v acetonitrile, 0.1% v/v formic acid, 1 µM yeast enolase peptide digest (Waters, 186002325) in MilliQ).

Tryptic peptides were analyzed on a Surveyor nano-LC system (Thermo) hyphenated to a LTQ-Orbitrap mass spectrometer (Thermo) (n = 1)[69]. Gold and carbon coated emitters (OD/ID = 360/25 µm tip ID = 5 µm), trap column (OD/ID = 360/100 µm packed with 25 mm robust Poros10R2/ 15 mm BioSphere $C_{18}$ 5 µm 120 Å) and analytical columns (OD/ID = 360/75 µm packed with 20 cm BioSphere $C_{18}$ 5 µm 120 Å) were from Nanoseparations (Nieuwkoop, The Netherlands). The mobile phases (A: 0.1% formic acid/$H_2O$, B: 0.1% formic acid/acetonitrile) were made with ULC/MS grade solvents (Biosolve). The emitter tip was coupled end-to-end with the analytical column via a 15 mm long TFE Teflon tubing sleeve (OD/ID 0.3 × 1.58 mm, Supelco, USA) and installed in a stainless steel holder mounted in a nano-source base (Upchurch scientific, Idex, USA). General mass spectrometric conditions were: electrospray voltage of 1.8 kV, no sheath and auxiliary gas flow, ion transfer tube temperature 150 °C, capillary voltage 41 V, tube lens voltage 150 V. Internal mass calibration was performed with polydimethylcyclosiloxane (m/z = 445.12002) and dioctyl phthalate ions

($m/z = 391.28429$) as lock mass. Samples of 10 μl were separated via a trap-elute setup at 250 nL/min flow and analyzed by data-dependent acquisition of one full scan/ top 3 method. For shotgun proteomics, fragmented precursor ions measured twice within 10 s were dynamically excluded for 60 s and ions with $z < 2$ or unassigned charges were not analyzed. Peptide ID was determined with the Mascot search engine. A parent ion list of the m/z ratios of the active-site peptides was compiled and used for LC-MS/MS analysis in a data-dependent protocol. The parent ion ($m/z$ $[M + 2H]^{2+} = 689.2671$) was electrostatically isolated in the ion trap of the LTQ and fragmented by MS/MS. Precursor ion was not observed in vehicle-treated control. Data from MS/MS experiments were validated manually.

**Inhibitor washout by dialysis**. FES$^{S700C}$-overexpressing HEK293T lysate (396 μL, 1.43 mg/mL) was incubated with vehicle, TAE684 or WEL028 (4 μL of 100x concentrated stock in DMSO) at final concentrations corresponding to their respective IC$_{80}$ values (TAE684: 225 nM, WEL028: 231 nM) for 30 min at rt. One fraction (200 μL) of the sample was immediately flash-frozen and stored at −80 °C until use. The remaining sample was transferred to a dialysis cassette (Slide-A-Lyzer™ Dialysis Cassette, 7K MWCO, 0.5 mL; Thermo Fisher), followed by dialysis in 200 mL PBS overnight at 4 °C. Pre- and post-dialysis samples were then incubated with probe WEL033 as described in the section "One-step probe labeling experiments" below, or conjugated to BODIPY-N$_3$ by CuAAC as described in the section "Two-step probe labeling experiments" below.

**Cell culture—general**. Cell lines were purchased at ATCC (HEK293T: ATCC® CRL-11268™, U2OS: ATCC® HTB-96™, HL-60: ATCC® CCL-240™) and were tested on regular basis for mycoplasma contamination. Cultures were discarded after 2–3 months of use. HEK293T (human embryonic kidney) and U2OS (human osteosarcoma) cells were cultured at 37 °C under 7% CO$_2$ in DMEM containing phenol red, stable glutamine, 10% (v/v) high iron newborn calf serum (Seradigm), penicillin and streptomycin (200 μg/mL each; Duchefa). Medium was refreshed every 2–3 days and cells were passaged two times a week at 80–90% confluence. HL-60 (human promyeloblast) cells were cultured at 37 °C under 5% CO$_2$ in HEPES-supplemented RPMI containing phenol red, stable glutamine, 10% (v/v) fetal calf serum (Biowest), penicillin and streptomycin (200 μg/mL each), unless stated otherwise. Cell density was maintained between $0.2 \times 10^6$ and $2.0 \times 10^6$ cells/mL. Cell viability was assessed by Trypan Blue exclusion and quantification using a TC20™ Automated Cell Counter (Bio-Rad).

**Cell culture—transfection**. One day prior to transfection, HEK293T or U2OS cells were transferred from confluent 10 cm dishes to 15 cm dishes. Before transfection, medium was refreshed (13 mL). A 3:1 (m/m) mixture of polyethyleneimine (PEI; 60 μg/dish) and plasmid DNA (20 μg/dish) was prepared in serum-free medium and incubated for 15 min at rt. The mixture was then dropwisely added to the cells, after which the cells were grown to confluence in 72 h (HEK293T) or 48 h (U2OS). Cells were then harvested by suspension in PBS, followed by centrifugation for 5 min at 200$g$. Cell pellets were flash-frozen in liquid nitrogen and stored at −80 °C until sample preparation.

Transfection of CRISPR plasmid DNA and ssODN repair template into HL-60 cells was performed using Amaxa nucleofector kit V and nucleofector I device (Lonza). One day prior to transfection, HL-60 cells were diluted to a density of $0.4 \times 10^6$ cells/mL. The next day, $2 \times 10^6$ cells per condition were centrifuged at 200$g$ for 5 min and resuspended in 100 μL nucleofection solution. Plasmid DNA (2 μg) and if applicable ssODN repair template (400 pmol) were added and cells were nucleofected using program T-019. Cells were allowed to recover for 10 min at rt before transfer to 12-well plates and further recovery in antibiotics-free medium at 37 °C.

**Cell culture—differentiation**. One day prior to induction of differentiation, cells were diluted to $0.4 \times 10^6$ cells/mL. Monocyte/macrophage differentiation was induced by addition of phorbol 12-myristate 13-acetate (PMA; 16 nM) for 48 h, during which cells attached to the plastic surface and acquired monocyte/macrophage morphology. Neutrophil differentiation was induced by addition of all-trans-retinoic acid (1 μM) and dimethylsulfoxide (1.25%) for 72–96 h. For morphological inspection, images were taken on an EVOS FL Auto 2 Imaging System (Thermo Fisher) at ×20 magnification.

**Cell culture—inhibitor treatment**. The term in situ is used to designate experiments in which live cell cultures are treated with inhibitor, whereas the term in vitro refers to experiments in which the inhibitor is incubated with cell lysates. Compounds were diluted in growth medium from a 1000x concentrated stock solution in DMSO. For in situ assays on live transfected cells, cells were transfected prior to treatment as described above. After 48 h, cells were treated with compound for 1 h. Cells were collected by suspension in PBS and centrifuged (1000$g$, 5 min, rt). Pellets were flash-frozen in liquid nitrogen and stored at −80 °C until use. For in situ treatment post-differentiation, HL-60 cells were differentiated as described above and incubated with compound for 1 h unless specified otherwise. Cells were collected by suspension for non-adherent cells and trypsinization for adherent cells. After collection, cells were centrifuged (200$g$, 5 min, rt) and washed in equal volume of PBS (1000$g$, 5 min, rt). Cell pellets were flash-frozen in liquid nitrogen and stored at −80 °C until use.

**CRISPR/Cas9 gene editing**. Selection of sgRNA was based on proximity to desired site of cleavage or mutagenesis as well as efficiency and specificity as predicted by CHOPCHOP v2 online web tool (http://chopchop.cbu.uib.no). sgRNA sequences can be found in Supplementary Table 7. For FES mutagenesis, a sgRNA targeting exon 16 was selected and an ssODN HDR donor was designed that after successful HDR incorporates the desired S700C mutation along with a BglII restriction site to facilitate analysis by RFLP. Furthermore, silent mutations are incorporated to remove "NGG" protospacer-adjacent motifs (PAMs), preventing recleavage of the genomic sequence after successful HDR or cleavage of the ssODN donor itself. For FES knockout, a sgRNA targeting exon 1 was selected.

For preparation of conditioned RPMI medium, HL-60 cells were diluted to $0.2 \times 10^6$ cells/mL and grown for 48 h at 37 °C. Cell suspension was then centrifuged (1000$g$, 5 min) and medium was transferred to a clean tube, followed by a second centrifugation step (3500$g$, 10 min). The medium was subsequently filtered through a 0.2 μm sterile filter and stored at −80 °C until use, for no longer than 3 months.

Single cell cultures were obtained approximately 7 days post-nucleofection by dilution in 1:1 conditioned and fresh RPMI medium and expanded in 96-well plates (100 μL per well). After 14 days, plates were inspected for cell growth, clones were collected in new plates and half of the volume was transferred to 96-well PCR-plates, followed by centrifugation (1000$g$, 10 min). Medium was removed and cell pellets were suspended in 25 μL QuickExtract™ (Epicentre). The samples were incubated at 65 °C for 6 min, mixed by vortexing and then incubated at 98 °C for 2 min. Genomic DNA extracts were diluted in sterile water and directly used in PCR reactions. Genomic PCR reactions were performed on 5 μL isolated genomic DNA extract using Phusion High-Fidelity DNA Polymerase (Thermo Fisher) in Phusion HF buffer in a final volume of 20 μL.

**Genotyping assays**. Restriction fragment length polymorphism (RFLP) assays were performed by combining genomic PCR product (8.5 μL) with FastDigest buffer (1 μL) and FastDigest BglII (0.5 μL; Thermo Fisher), followed by incubation at 37 °C for 30 min.

For T7E1 assays, genomic PCR products were mixed in a 1:1 ratio with WT amplicons and denatured and reannealed in a thermocycler using the following program: 5 min at 95 °C, 95 to 85 °C using a ramp rate of −2 °C/s, 85 to 25 °C using a ramp rate of −0.2 °C/s. To annealed PCR product (8.5 μL), NEB2 buffer (1 μL) and T7 endonuclease I (5 U, 0.5 μL; New England Biolabs) were added, followed by incubation at 37 °C for 30 min. Digested PCR products were analyzed using agarose gel electrophoresis with ethidium bromide staining.

For sequencing analysis, genomic PCR products were purified using NucleoSpin® Gel and PCR Clean-up kit (Macherey-Nagel), followed by Sanger sequencing. For knockouts, sequence traces were decomposed using the TIDE online web tool (www.deskgen.com). The efficiency for obtaining homozygous mutant clones was approximately 0.5–1% (typically 1 out of 100–200 clones), whereas the efficiency of obtaining full knockout clones was approximately 10%.

Potential off-target cleavage sites of the used sgRNA were predicted using DESKGEN™ online web tool (www.deskgen.com). Top-ranked potential coding off-target sequences containing three or less mismatches to the employed sgRNA sequence were selected for validation. The genomic region surrounding the potential off-target site was amplified by PCR and analyzed by Sanger sequencing.

**Preparation of cell lysates**. Pellets of HEK293T or U2OS cells were thawed on ice and suspended in lysis buffer (50 mM HEPES pH 7.2, 150 mM NaCl, 1 mM MgCl$_2$, 0.1% (w/v) Triton X-100, 2 mM Na$_3$VO$_4$, 20 mM NaF, 1 x mammalian protease inhibitor cocktail, 25 U/mL benzonase). Cells were lysed by sonication on ice (15 cycles of 4″ on, 9.9″ off at 25% maximum amplitude). Protein concentration was determined using Quick Start™ Bradford Protein Assay (Bio-Rad) and diluted to appropriate concentration in dilution buffer (50 mM HEPES pH 7.2, 150 mM NaCl). Lysates were aliquoted, flash-frozen and stored at −80 °C until use.

Pellets of HL-60 cells were thawed on ice and suspended in M-PER buffer supplemented with 1x Halt™ phosphatase and protease inhibitor cocktail (Thermo Fisher), followed by centrifugation (14,000$g$, 10 min, 4 °C). The resulting clear lysate was further processed as described above.

**One-step probe labeling experiments**. For in vitro inhibition experiments, cell lysate (14 μL) was preincubated with inhibitor (0.5 μL, 29 × concentrated stock in DMSO, 30 min, rt), followed by incubation with WEL033 (0.5 μL, 30 × concentrated stock in DMSO, 30 min, rt). For in situ inhibition experiments, treated cell lysate (14.5 μL) was directly incubated with WEL033 (0.5 μL, 30 × concentrated stock in DMSO, 30 min, rt). Final concentrations of inhibitors and/or WEL033 are indicated in the main text and figure legends. Reactions were quenched with 4× Laemmli buffer (5 μL, final concentrations 60 mM Tris pH 6.8, 2% (w/v) SDS, 10% (v/v) glycerol, 5% (v/v) β-mercaptoethanol, 0.01% (v/v) bromophenol blue) and boiled for 5 min at 95 °C. Samples were resolved on 10% acrylamide SDS-PAGE gel (180 V, 75 min). Gels were scanned using Cy3 and Cy5 multichannel settings (605/ 50 and 695/55 filters, respectively; ChemiDoc™ MP System, Bio-Rad). Fluorescence intensity was corrected for protein loading determined by Coomassie Brilliant Blue

R-250 staining and quantified with Image Lab (Bio-Rad). $IC_{50}$ curves were fitted with Graphpad Prism® 7/8 (Graphpad Software Inc.).

**Two-step probe labeling experiments**. For in vitro experiments, cell lysate (12 μL) was preincubated with inhibitor (0.5 μL, 25× concentrated stock in DMSO, 30 min, rt), followed by incubation with WEL028 (0.5 μL, 26× concentrated stock in DMSO, 30 min, rt). Meanwhile, "click mix" was prepared freshly by combining $CuSO_4$ (1 μL of 15 mM stock), sodium ascorbate (0.6 μL of 150 mM stock), THPTA (0.2 μL of 15 mM stock) and fluorophore-azide (0.2 μL of 150× concentrated stock in DMSO, 2 eq.). Click mix was added to the reaction, followed by incubation for 30 min at rt, after which the reaction was quenched and further processed as described above. For in situ inhibition experiments, WEL028-treated cells were lysed and directly incubated with click mix.

**Chemical proteomics**. Cell lysates of FES$^{WT}$ and FES$^{S700C}$ HL-60 macrophages or neutrophils incubated in situ with WEL028 or vehicle (as negative control) were prepared as aforementioned ($n = 3$ or 4 biological replicates). Lysates (250 μL of 2 mg/mL) were conjugated to biotin-azide using click chemistry (25 μL click mix, 1 h, 37 °C). The reaction was quenched and excess biotin-azide was removed by chloroform/methanol precipitation. Precipitated proteome was suspended in 6 M urea in 25 mM ammonium bicarbonate, reduced (10 mM DTT, 15 min, 65 °C) and alkylated (40 mM iodoacetamide, 30 min, rt, in the dark). SDS was added (2% final concentration, 5 min, 65 °C), samples were diluted in PBS and incubated with avidin beads (from 50% slurry, 3 h, rt, in overhead rotator). Beads were washed with 0.5% SDS in PBS, followed by 3 washes with PBS, and then transferred to low-binding Eppendorf tubes. Proteins were digested with trypsin overnight at 37 °C and resulting peptides were desalted using stage tips with $C_{18}$ material as described in the section "MS identification of covalent inhibitor-peptide adduct" above.

The peptides were measured on a NanoACQUITY UPLC System coupled to SYNAPT G2-Si high definition mass spectrometer[69]. A trap-elute protocol, where 5 μL of the digest was loaded on a trap column ($C_{18}$ 100 Å, 5 μm, 180 μm × 20 mm, Waters) followed by elution and separation on the analytical column (HSS-T3 $C_{18}$ 1.8 μm, 75 μm × 250 mm, Waters). The sample was brought onto this column at a flow rate of 10 μL/min with 99.5% solvent A (0.1% formic acid/$H_2O$) for 2 min before switching to the analytical column. Peptide separation was achieved using a multistep concave gradient. The column was re-equilibrated to initial conditions after washing with 90% solvent B (0.1% formic acid/acetonitrile). The rear seals of the pump were flushed every 30 min with 10% (v/v) acetonitrile. [Glu1]-fibrinopeptide B (GluFib) was used as a lock mass compound. The auxiliary pump of the LC system was used to deliver this peptide to the reference sprayer (0.2 μL/min). The mass range was set from 50 to 2000 Da with a scan time of 0.6 s in positive, resolution mode. The collision energy was set to 4 V in the trap cell for low-energy MS mode. For the elevated energy scan, the transfer cell collision energy was ramped using drift-time specific collision energies. The lock mass was sampled every 30 s. For raw data processing, PLGS (version 3.0.3) was used. The MS$^E$ identification workflow was performed to search the human proteome from Uniprot (uniprot-homo-sapiens-trypsin-reviewed-2016_08_29.fasta). Protein quantification was performed using ISOQuant (version 1.5). Relevant data processing parameters are found in Supplementary Table 8 and 9[69].

The following cut-offs were used for target identification: unique peptides ≥ 1, identified peptides ≥ 2, ratio WEL028-treated over vehicle-treated ≥ 2 with q-value < 0.05 based on t-test with Benjamini-Hochberg multiple comparison correction (FDR = 10%), kinase annotation in Uniprot database.

**Immunoblot**. Samples were resolved by SDS-PAGE as described above, but transferred to 0.2 μm polyvinylidene difluoride membranes by Trans-Blot Turbo™ Transfer system (Bio-Rad) directly after fluorescence scanning. Membranes were washed with TBS (50 mM Tris pH 7.5, 150 mM NaCl) and blocked with 5% milk in TBS-T (50 mM Tris pH 7.5, 150 mM NaCl, 0.05% Tween-20) for 1 h at rt.

Membranes were then incubated with primary antibody in 5% milk in TBS-T (FLAG, V5, β-actin; o/n at 4 °C) or washed three times with TBS-T, followed by incubation with primary antibody in 5% BSA in TBS-T (other antibodies, o/n at 4 °C). Membranes were washed three times with TBS-T, incubated with matching secondary antibody in 5% milk in TBS-T (1:5000, 1 h at rt) and then washed three times with TBS-T and once with TBS. Luminol development solution (10 mL of 1.4 mM luminol in 100 mM Tris pH 8.8 + 100 μL of 6.7 mM p-coumaric acid in DMSO + 3 μL of 30% (v/v) $H_2O_2$) or Clarity Max™ ECL Substrate (Bio-Rad) was added and chemiluminescence was detected on ChemiDoc™ MP System.

Following detection of phospho-proteins, membranes were stripped (Restore™ Plus Stripping Buffer, Thermo Fisher) for 20 min, washed three times with TBS, and blocked and incubated with the control antibodies as described above.

Primary antibodies: monoclonal mouse anti-FLAG M2 (1:5000, Sigma Aldrich, F3156), monoclonal anti-V5 (1:5000, Thermo Fisher, R960–25), monoclonal mouse anti-β-actin (1:1000, Abcam, ab8227), polyclonal rabbit anti-Lamin B1 (1:5000, Thermo Fisher, PA5-19468), polyclonal rabbit anti-phospho-FES Y713 (1:1000, Thermo Fisher, PA5-64504), monoclonal rabbit anti-FES (1:1000, Cell Signaling Technology (CST), #85704), polyclonal rabbit anti-phospho-SYK Y352 (1:1000, CST, #2701), monoclonal rabbit anti-SYK (1:1000, CST, #13198), polyclonal rabbit anti-phospho-HS1 Y397 (1:1000, CST, #4507), polyclonal rabbit

anti-HS1 (1:1000, CST, #4503), polyclonal rabbit anti-phospho-PLCγ2 Y1217 (1:1000, CST, #3871), polyclonal rabbit anti-PLCγ2 (1:1000, CST, #3872). Secondary antibodies: goat anti-mouse-HRP (1:5000, Santa Cruz, sc-2005), goat anti-rabbit-HRP (1:5000, Santa Cruz, sc-2030).

**CD11b expression analysis by flow cytometry**. Cells ($1 \times 10^6$ per sample) were centrifuged (500g, 3 min) and suspended in human FcR blocking solution (Miltenyi Biotec, 25× diluted in FACS buffer (1% BSA, 1% FCS, 0.1% $NaN_3$, 2 mM EDTA in PBS)), transferred to a V-bottom 96-well plate and incubated for 10 min at 4 °C. Next, monoclonal rat CD11b-APC antibody (1:100, Miltenyi Biotec, 130-113-231) or rat anti-IgG2b-APC isotype control antibody (1:100, Miltenyi Biotec, 130-106-728) was added along with 7-AAD (1 μg/mL) and samples were incubated for 30 min 4 °C in the dark. Samples were washed once in PBS and fixed in 1% PFA in PBS for 15 min at 4 °C in the dark, followed by two washing steps in PBS and resuspension in FACS buffer to a density of approximately 500 cells/μL. Cell suspensions were measured on a Guava easyCyte HT and data was processed using GuavaSoft InCyte 3.3 (Merck Millipore). Events (generally 20,000 per condition) were gated by forward and side scatter (cells), side scatter area (singlets) and viability (live cells) and the percentage of CD11b-positive cells was determined based on background fluorescence for isotype control antibody and non-differentiated cells. The RED-R channel (661/15 filter) and RED-B channel (695/50 filter) were used to detect CD11b-APC and 7-AAD, respectively.

**Neutrophil oxidative burst assay**. Cells were centrifuged (500g, 3 min), suspended in PBS and seeded in triplicate per condition in 96-well plates (100,000 cells per well in 100 μL). To each well, 100 μL of PBS containing nitroblue tetrazolium (NBT) with vehicle or PMA was added, bringing final concentrations to 0.1% and 1.6 μM, respectively. Cells were incubated (1 h, 37 °C) and plates were imaged by phase contrast microscopy (20x magnification, EVOS FL Auto 2). Cells positive for formazan deposits were counted (three different fields per replicate).

**TempO-Seq transcriptome profiling**. HL-60 cells were seeded at 20,000 cells per well in a 96-well plate and differentiated towards macrophages as aforementioned with three independent biological replicates. After 48 h, medium was removed, cells were washed with 100 μL PBS and subsequently lysed in 50 μL 1x BioSpyder lysis buffer (15 min, rt). Lysates were flash-frozen in liquid nitrogen, stored at −80 °C and shipped to BioSpyder technologies on dry ice, where the TempO-Seq® profiling was conducted according to supplier's standard procedures[48]. Raw gene transcription counts were subjected to internal normalization using DESeq2 and statistical significance determined by t-test with Benjamini-Hochberg multiple comparison correction (FDR = 1%). Differentially expressed genes were identified using following cut-offs: fold change of corrected counts of FES$^{S700C}$ over WT HL-60 cells < 0.5 or >2 with q-value < 0.05.

**Co-immunoprecipitation**. U2OS cells were co-transfected with FLAG-tagged FES and V5-tagged SYK and grown for 48 h as described above, followed by incubation with vehicle or WEL028 (200 nM) for 1 h. Cells were then washed with PBS and collected by scraping in IP-buffer (20 mM Tris-HCl pH 7.5, 150 mM NaCl, 1% Triton X-100 supplemented with 1g Halt™ phosphatase and protease inhibitor cocktail (Thermo Fisher)). Cells were lysed by sonication on ice (three cycles of 10″ on, 10″ off at 25% maximum amplitude), centrifuged (14,000g, 10 min, 4 °C). The clear lysate was diluted to 1 mg/mL in IP-buffer and subjected to immunoprecipitation using Dynabeads™ Protein G Immunoprecipitation kit (Thermo Fisher) following manufacturer's protocol. Briefly, anti-FLAG M2 antibody (1:100, Sigma Aldrich, F3156) was incubated with beads with gentle rotation (10 min, rt), after which lysate (500 μL of 1 mg/mL) was added and incubated (1 h, 4 °C). Beads were washed three times, transferred to clean tubes and eluted by suspension in 2× Laemmli buffer (50 μL, 10 min, 70 °C). Samples (10 μL per lane) were resolved by SDS-PAGE and immunoblotted using anti-V5 or anti-FLAG antibodies. Immunoprecipitations were performed in three independent replicates.

**Phagocytosis assay**. HL-60 cells were differentiated to neutrophils as described above. Cells were counted and centrifuged (200g, 5 min), followed by resuspension in growth medium without antibiotics. Cells were incubated with vehicle or inhibitor (from 1000x concentrated stocks in DMSO) for 1 h at 37 °C prior to infection ($1\times10^6$ cells in 900 μL in 12-well plate). Meanwhile, E. coli B834(DE3) constitutively expressing GFP$^{A206K}$ were grown in LB medium to an OD600 of 0.4–1.0, after which bacteria were centrifuged (2000g, 5 min), washed and resuspended in PBS to appropriate density[70]. Neutrophils were infected by addition of bacteria at multiplicity of infection (MOI) of 30 unless stated otherwise. Cells were then incubated for 1 h at 37 °C unless stated otherwise, after which cells were resuspended and transferred to Eppendorf tubes, washed in FACS buffer (1 mL, 500g, 3 min) and fixed in 1% PFA in PBS (15 min, 4 °C, in the dark). Samples were further processed as described above.

Events (generally 20,000 per condition) were gated by forward and side scatter (cells), side scatter area (singlets) and the percentage of GFP-positive cells and GFP mean fluorescence intensity (MFI) were determined based on background fluorescence for non-infected cells. The GREEN-B channel (525/30 filter) was used to detect GFP. Phagocytic index was calculated as fraction of GFP-positive cells

(number of phagocytic cells) multiplied by GFP MFI (number of phagocytized bacteria).

**Neutrophil stimulation and phosphoprotein analysis**. HL-60 neutrophils were infected as described above, but in 1.5 mL tubes ($5 \times 10^6$ cells per sample, in 1 mL growth medium without antibiotics, supplemented with FcR blocking reagent (1:25)). After indicated infection times, suspensions were immediately centrifuged (2000g, 2 min, 4 °C). Supernatant was completely aspirated and cells were washed (1 mL cold PBS, 2000g, 2 min, 4 °C). Pellets were thoroughly resuspended in 1× Laemmli sample buffer (75 µL), followed by incubation at 95 °C (15 min, 1400 rpm) and brief sonication to reduce sample viscosity. Samples were stored at −20 °C or immediately resolved by SDS-PAGE.

**Statistics and reproducibility**. All statistical measures and methods are included in the respective figure or table captions. In brief: all replicates represent biological replicates and all data represent means ± SEM, unless indicated otherwise. Statistical significance was determined using Student's t tests (two-tailed, unpaired) or ANOVA with Holm-Sidak's multiple comparisons correction. ***$P < 0.001$; **$P < 0.01$; *$P < 0.05$; NS if $P > 0.05$. All statistical analyses were conducted using GraphPad Prism® 7/8 or Microsoft Excel.

Reproducibility of experiments was confirmed by the use of separately measured (biological) replicates and/or appropriate controls. In vitro biochemical and cellular experiments (including results presented in Figs. 4b, c, 5c–h and Supplementary Figs. 6a–c, 7a and 9h) were performed at least in two independent experiments, yielding similar results. For quantified data, the exact number of replicates per data point is indicated in figure legends.

**Reporting summary**. Further information on research design is available in the Nature Research Reporting Summary linked to this article.

## Data availability
The source data underlying Figs. 2–7, Table 1, Supplementary Figs. 1–2, 4–9 and 12–18 and Supplementary Tables 2–4 are provided as a Source Data file. Other data are available from authors upon request.

The TempO-Seq data associated with Fig. 5g are available online in the Gene Expression Omnibus (GEO) database (https://www.ncbi.nlm.nih.gov). The GEO accession number is GSE145811.

The proteomics data associated with Fig. 6f-g and Fig. 7c are available online in the Proteomics Identifications Database (PRIDE) (https://www.ebi.ac.uk/pride). The PRIDE dataset identifier is PXD018270. Substrate consensus motif was generated using Enologos (http://www.benoslab.pitt.edu). The DESKGEN™ (www.deskgen.com) and CHOPCHOP v2 (http://chopchop.cbu.uib.no) online web tools were used for CRISPR design and analyses. Uniprot databases were used for proteomics analyses (https://www.uniprot.org/proteomes). Source data are provided with this paper.

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

## Acknowledgements

The Cloning & Purification Facility and Hans van den Elst are acknowledged for technical support. We thank Richard J.B.H.N. van den Berg for critically reading the Material & Methods and Jessica Schlachter for preparing the PAK4 plasmids. Prof. dr. John Kuriyan is acknowledged for providing the YopH plasmid. The department of Toxicology (Leiden Academic Centre for Drug Research) is kindly acknowledged for access to the TempO-Seq consortium. We acknowledge ChemAxon for kindly providing the Instant JChem software to manage our compound library. Robert van Sluis is kindly acknowledged for graphic illustrations. T.v.d.W., A.K, T.B., and M.v.d.S. would like to acknowledge NWO (VICI) & Topsector Chemistry (TKI-project "OncoDrugs") for financial support. E.B.L. would like to thank the Agentschap Innoveren en Ondernemen (AIO) for financial support (AIO project 155028). G.J.P.v.W. thanks the Dutch Research Council (NWO-TTW Veni 14410) for funding.

## Author contributions

T.v.d.W., A.K., T.B., and M.v.d.S. designed research approach. T.v.d.W., H.d.D., R.H., T.v.d.H., N.M.P., J.A.P.M.W, B.I.F., and E.B.L. performed experiments and analyzed data. T.v.d.W. and M.v.d.S. wrote the manuscript. R.R., G.J.P.v.W. H.S.O., A.K., and T.B. provided useful comments and feedback to improve the manuscript.

## Competing interests

A.K. and T.B. are owners of Covalution Biosciences BV. R.H. is an employee of Pam-Gene, T.v.d.H. and R.R. were employees of PamGene at the time of the research. The authors declare no competing interests.
