## [Peer Review File · Nature Communications]

Reviewers' comments:

Reviewer #1 (Remarks to the Author):

This manuscript presented by van der Wel et al. features a demonstration of the efficacy and applicability of a chemical-genetic strategy in studying kinase function and target engagement. CRISPR/Cas9 gene editing was employed to introduce a cysteine mutation at the DFG-1 position in endogenously expressed FES kinase, for which complementary, covalent inhibitors were synthesized. Engagement of mutant FES by its cognate inhibitor allows for temporally controlled modulation of catalytic activity, thereby enabling the minimally-invasive study (previous reports rely on transient/stable overexpression or knock-out models) of its physiological role. Acute inactivation of mutant FES revealed two insights in HL-60 cells: 1) FES activity is dispensable during differentiation along the macrophage lineage and 2) SYK is a downstream target of FES, for which its phosphorylation regulates bacterial phagocytosis in mature neutrophils. Furthermore, mutant-specific labeling was achieved with three other kinases – LYN, PTK2, and PAK4 – after introducing a DFG-1 cysteine, suggesting the broad amenability of this position as a handle for covalent probe engagement.

Although the strategy employed in the present study is not entirely novel (mutagenic introduction of a cysteine to enhance potency and selectivity of an inhibitor has been used to study various other kinases: Poulidakos, 2010; Kung, 2017; etc.), this study successfully applies the chemical-genetic approach with endogenous levels of the kinase-of-interest and identified a new site amenable such a strategy. The importance of such a study is quite significant, especially considering recent reports detailing confounding results in overexpression and knock-out models. Moreover, extensive characterization of the mutants is performed, ensuring orthogonality of the cysteine mutation in HL-60 cell physiology. As such, this reviewer recommends conditional acceptance of the manuscript for publication in Nature Communications, contingent upon addressing several minor/moderate concerns.

Specific comments:

1. SelectScreen™ screening of 279 kinases was performed to assess kinome-selectivity of WEL028 (supplementary figure 3). Based on the language in-text, it is slightly unclear what the selection criteria was for the kinases included in the screen. Do the 279 kinases contain “native cysteine residues at any position in the active site” (page 6) or is there a smaller sub-set of the 279 that contain such a cysteine? If the latter, why were these particular kinases selected for the screen and why were the others excluded?

2. Can the authors justify the need for including both a one-step (WEL033) and a two-step (WEL028) fluorescent probe in their study? Rather than incorporate the fluorescent tag directly on the

inhibitor (and thus face cell permeability issues, fluorescent decay after extended storage, etc.) the authors could easily perform the CuAAC reaction with Cy5-azide after covalent labeling of WEL028 on FES targets. Another alternative strategy to the target engagement competition experiments performed (in which pre-treatment with WEL028 is followed by treatment with WEL033) is to pre-treat cells, protein, etc. with inhibitor 5 and subsequently treat with WEL028, followed by Cy5-azide CuAAC. Is there a particular reason for performing one-step probe labeling experiments?

3. Supplemental figure 5b appears to have mislabeled lanes. As labeled, pre-treatment with 4, followed by WEL028 treatment appears to lead to an intense fluorescent signal whereas WEL028 alone does not. Perhaps these samples were swapped during gel loading? Otherwise, the data seems to be irreconcilable with the rest of the data in the manuscript.

4. In order to determine an irreversible, covalent mode of binding for WEL028, inhibitor washout experiments were performed (supplementary figure 6d,e). It is slightly concerning that the vehicle-treated sample post-dialysis leads to such a weak fluorescent signal. There is no obvious reason why post-dialysis intensity should be so much lower, despite lower protein levels in general. This is likely leading to an artificially low signal for WEL028-treated samples and an artificially high signal for TAE684-treated samples post-dialysis (considering normalization to vehicle). If the authors are satisfied with the quality of the data, how do they rationalize the relatively dramatic increase in fluorescence for TAE684-treated samples post-dialysis? Surely TAE684 isn't enhancing affinity for WEL033.

5. To ensure that FESS700C HL-60 cells differentiate in a manner identical to FESWT cells, proliferation and CD11b-receptor expression between the two alleles were monitored and compared (Fig 4e,f). The text contains mis-references (bottom of page 8 and top of page 9) to the figures: fig 4e contains cell proliferation data while fig 4f describes CD11b expression percentages.

6. The authors provide very little discussion regarding the presence of several off-targets (Fig 4g) besides a brief acknowledgement of their existence (page 9). The kinome screen performed revealed a few kinases (LRRK2, ~286 kDa; MKNK2, 46 kDa) for which WEL028 is only roughly 10-fold less selective against. The chemical proteomic analysis revealed additional ones (MAPK1/MAP2K2/MAP2K1/GAK etc.). Under the conditions described (1 μ M WEL033), is there any concern that the off-targets could potentially complicate downstream biological characterization? The authors should discuss this important point.

7. A targeted transcriptomics analysis was performed using TempO-Seq to ensure that FESS700C HL-60 cells exhibit few to no transcriptional deviations during macrophage differentiation (Fig 5a). The authors conclude that the mutation minimally disturbs gene expression based on the presence of

only 7 identified transcripts that are altered in mutant FES cells. While this appears rational from a quantitative perspective – 7 out of ~21,000 total transcripts is a small, insignificant percentage (0.03%) – there is no characterization of the qualitative effect of these alterations, albeit few. It seems perfectly plausible that even if only 1 altered transcript is identified, its alteration could have significant effects beyond the transcriptome level.

Reviewer #2 (Remarks to the Author):

Van der Wel and colleagues have engineered a FES kinase mutant with residue Ser700 near the activation loop changed from Ser to Cys. They then modified the TAE684 small molecule FES inhibitor that had previously been co-crystallized with FES (Hellwig et al. 2012) to find derivatives that were more selective inhibitors of FES-S700C, relative to FES-WT. Optimized molecules were further modified to include an electrophilic group that would form a covalent bond through a Michael addition to the novel thiol at C700. Variant of this compound were generated with a fluorescent group to facilitate labelling FES-S700C or biotin to enable affinity purification.

Biochemical analysis indicates these compounds effectively and differentially inhibit FES S700C, relative to FES WT. However, the specificity of these compounds with respect to the kinome is overstated as they still inhibited 50% of the activity of 19 of 279 kinases profiled at 1 μ M. By extension, one would expect ~38 of the human 518 kinases to be significantly inhibited at 1 μ M concentration. This degree of specificity is comparable with a wide range of existing kinase-“specific” inhibitors. Thiol directed covalently acting kinase inhibitors are not novel; and this set of inhibitors are not particularly specific for the FES-S700C target.

The rationale for developing this thiol-directed inhibitor for an engineered FES kinase was confusing because the authors discussed it in the context of ATP analog sensitive kinase derivatives used by Shokat and colleagues to identify kinase substrates. In the Shokat approach, it is the gatekeeper hinge region of the kinase that is altered to allow derivative forms of ATP to be used as substrates to selectively phosphorylate and thereby identify kinase substrates. In contrast, the work described here by Van der Wel and colleagues is directed at selectively inhibiting and covalently labeling an engineered FES variant. Unfortunately, the compounds they developed still inhibit and label many other kinases/targets. The extent to which these reagents are selective towards the engineered FES-S600C variant is overstated.

CRISPR/Cas9 methods were then used to engineer the same S700C substitution into the endogenous FES locus in HL60 cells, a human promyelocytic leukemia cell line that can be induced to differentiate into granulocytes or macrophages. The rationale for this was to be able to explore the effects of selectively inhibiting FES on differentiation, proliferation, bacterial phagocytosis and signaling in a physiologically relevant context. They make an argument that this will be more informative than a genetic approach because that is prone to adaptation. This argument is also overstated because cells can also adapt to drug challenges; but of more concern, their inhibitors may act on other kinases (or other targets) in this system whereas genetic approaches are more likely to be specific. Surprisingly, the authors did not generate a simple CRISPR/Cas9 FES knockout HL60 cell line and use it to perform a head to head comparison.

CD11b expression in PMA-induced HL60 cells was used to assess myeloid differentiation. These data (Fig 5d,e) would seem to indicate that FES suppresses differentiation. However, statistics were not shown in the case of the data in Figure 5e, where WT and FES-S600C cells were compared. The greater effect on myeloid differentiation at 1uM was interpreted as due to off-target effects. Indeed, the labeling experiments in Fig 4g and Fig 5b indicate that FES-S700C represents a minority of the proteins that are tagged with their inhibitor.

The authors next show data indicating that FES-S700C inhibition with WEL028 can reduce bacterial phagocytosis. (Fig 6d,e). It was not clear how they quantified the data in Fig 5d to arrive at the phagocytic indices in the graphs shown in Fig 5e,i. In Fig 5e, where WT and FES-S600C cells are compared, the effect looks to be ~25%. This is consistent with the 36% reduction in bacterial phagocytosis that was previously reported in FES knockout mouse resident peritoneal macrophages relative to WT (Parsons and Greer, 2006. *J Leukoc Biol.* 80:1522). Van der Wel and colleagues cite this paper a couple times in their manuscript, but do not state that a role for FES in bacterial phagocytosis was previously demonstrated in that study. Other previous studies using mice targeted with either null or kinase-inactivating mutations in *Fes* have also demonstrated that FES is not required for hematopoiesis (Senis et al 1999. *Mol Cell Biol* 19:7436; Senis et al 2003. *Exp Hematol.* 31:673), including the generation of granulocytes and macrophages; and one of these studies also explored SYK, PLC γ 2 and HS1 tyrosine phosphorylation during platelet activation and saw no effects with FES genetic disruption (Senis et al 2003. *J. Thrombosis and Haemostasis* 5:1062).

Van der Wel and colleagues show some evidence for a physical association between FES and SYK in a co-overexpression co-IP experiment (Fig 6h). Adding the reciprocal co-IP would have made this more compelling. They also show some evidence for FES mediated phosphorylation/activation of SYK; again in a co-overexpression system. SYK pY352 was abolished by WEL028, and this was interpreted as evidence for FES mediated SYK activation. However, in Sup Fig 11 it is apparent that this SYK pY352 signal is also seen in the absence of co-transfected FES, so it seems very likely that SYK can autophosphorylate under these over-expression conditions. It is difficult to reconcile these two pieces of data (Fig 6g and Sup Fig 11). How is it that SYK pY352 is inhibited by WEL028 in the presence of co-overexpressed FES-S700C, but not in its absence?

They then explore signaling in HL60 FES-S700C cells that have apparently been differentiated into neutrophils (although it is not specifically stated that these cells were treated with ATRA/DMSO). SYK, PLCg2 and HS1 phosphorylation was observed after incubation with bacteria, and this was inhibited by WEL028. Surprisingly, this experiment was not done on control HL60 cells in parallel. In the absence of this control, it is not valid to conclude that the loss of phosphorylation of SYK, PLCg2 and HS1 in the presence of WEL028 is due to FES inhibition. Furthermore, they provide no evidence that FES is activated under these conditions of bacterial challenge.

The final figure provides a model for FES recruitment to sites of bacterial phagocytosis and downstream activation of SYK and subsequent activation of PLCg2 and HS1. This is a highly speculative model with very little evidence in the manuscript to support it.

Other points:

The PamChip peptide array-based comparison of FES-WT vs FES-S700C activity looks impressive (Fig 2e), but it would be more compelling if a different kinase was included as a control. How many of the peptides on this array are validated FES targets? The same issue applies to the SH2 binding PamChip analysis (Fig 2g); and in this case, there do appear to be some significant differences that the authors don't acknowledge.

In Figure 4g the presumptive 93 kDa FES band is one of several species labeled with 1 μ M WEL033 in HL60 cells. FES is a minor species, likely in the range of <10% of the labeled proteins. This speaks to the lack of specificity of this probe.

Reviewer #3 (Remarks to the Author):

This study reports the development of a chemical genetics strategy to probe the role of the c-Fes non-receptor tyrosine kinase in macrophage differentiation and neutrophil function. The approach involves a single amino acid substitution of Ser700 with Cys in the kinase domain, which in turn enables irreversible inhibition with an electrophilic compound based on TAE844, a competitive inhibitor previously reported to inhibit Fes. The mutant kinase is characterized extensively in vitro, and then engineered into the genome of the human HL-60 promyelocytic leukemia cell line using CRISPR technology to avoid artefacts related to over-expression. Using the modified cells and probe compound, the authors show that Fes activity is not required for induction of macrophages from HL-

60 cells following phorbol ester treatment. However, they do show a role for Fes activity in phagocytosis of bacteria by neutrophils derived from HL-60s, through a pathway linked to activation of Syk.

Overall this is a very impressive study that effectively combines many aspects of chemical biology and mass spectrometry in support of the ultimate conclusions. The manuscript is very well written, and the data are presented in a clear and logical manner. Below are a few questions and comments for the authors to consider:

1. In the Introduction (page 2), the authors state that Fes knock-out mice revealed a role for Fes in myeloid differentiation. In fact, the opposite is true - Fes homozygous-null animals did not show any defect in myelopoiesis.

2. Results, page 5: In the text and in Table 1, please mention the concentration of ATP used for the kinase inhibition assays, and the relationship of the ATP concentration to the K_m for each kinase.

3. Regarding the experiments with HL-60 cells, it would be very helpful to the reader to include in the main text and the figure legends more details of the differentiation protocols. Please specify the agents used to induce differentiation to macrophages vs. neutrophils along with the concentrations used and the period of time required for differentiation induction. Also no phenotypic or morphological data is included to verify the extent of neutrophil differentiation in response to ATRA or DMSO.

4. In Figure 4, panels e and f should be reversed.

5. In Figure 5b, it is clear that treatment with WEL028 selectively engages Fes S700C vs. WT Fes. However, does this correlate with complete suppression of Fes kinase activity? This could be confirmed by independent IP of Fes followed by immunoblotting with the pY713 antibody.

6. Figure 7h shows a very powerful control, in which kinases labeled with WEL028 at a high concentration (1,000 nM) are captured and identified by MS to define the off-targets. Has this experiment been performed at 100 nM? Such a result, showing that only Fes is captured at the lower concentration, would provide very strong evidence for on-target performance of this compound at an appropriate concentration.

7. Data presented suggest that FES activity is not necessary for PMA-induced differentiation of HL-60 cells to macrophages. Did the authors test whether HL-60 differentiation along the neutrophil pathway via all-trans retinoic acid or DMSO treatment also requires FES activity? Early studies established that Fes activity is induced by these agents, but whether its activity is required for differentiation is not clear.

Reviewer #4 (Remarks to the Author):

In the manuscript “Chemical genetics strategy to profile kinase target engagement reveals role of FES in neutrophil phagocytosis via SYK activation” from the van der Stelt lab, the authors designed a mutant specific probe for the FES kinase and used these tools to understand the role of this kinase in downstream signaling pathways including myeloid differentiation and endocytosis.

The paper is interesting. By introducing a Cys in close proximity to highly conserved DFG motif, they could ablate the function the FES kinase with covalent interaction of probe and further investigated the downstream signaling pathway. Using competition assays, rational design and biochemical assays the authors demonstrated the specific interaction of probe with FES kinase. Moreover, by acute inhibition of FES, they show that FES is dispensable for differentiation of myeloid lineage; instead, they show this kinase has a role in receptor mediated endocytosis of *E. coli*. They showed that in this process the FES protein become auto phosphorylated and induced phosphorylation of other downstream kinase such as PLC gamma2 and HS1. Overall, the paper is rational and tried to use both genetic and chemoproteomic approaches to understand the role of FES as an important kinase involved in cancer and immune disorders.

Major point:

Here, the author tries to map the probe binding site and see that besides FES, several other proteins can be inhibited with the probe (the probe is not specific for only FES kinase). As shown in the data, the probe can interact with other cysteines close to the DFG motif. This results in a lot of complexity to the signaling pathway downstream of their specific inhibitor and understanding the function of FES. The result shows convincing data that the role of FES is not important in the process of myeloid differentiation.

As next step, the role of FES kinase has been studied in the context of endocytosis process and phosphorylation of several targets shown to be disrupted upon probe treatment. Here, a critical control is missing due to the fact that this might be due to the off-target activity of the probe.

The following should be addressed:

1. The authors should include both control and CRISPR modified cell line treated with inhibitor (with different concentrations to establish dose dependence) to see whether the signalling pathway can be regulated differently (the phosphorylation events upon E coli infection).
2. Furthermore, the study of signalling pathways in a FES knockout cell line can be considered as further support of the role of FES kinase in endocytosis.
3. Moreover, it would be interesting to see whether the mutation in tyrosine 713 in FES kinase or dominant negative form of the kinase lead to the same defect in the signalling pathway.

Point-by-point reply for manuscript “Chemical genetics strategy to profile kinase target engagement reveals role of FES in neutrophil phagocytosis via SYK activation” (NCOMMS-19-38211).

Please find below a point-by-point reply (R) to the reviewers' comments (C), followed by any revisions that were made in the manuscripts. We appreciate the time invested by the reviewers in evaluating our manuscript and thank them for their valuable comments and insights.

Reviewer #1 (Remarks to the Author):

This manuscript presented by van der Wel et al. features a demonstration of the efficacy and applicability of a chemical-genetic strategy in studying kinase function and target engagement. CRISPR/Cas9 gene editing was employed to introduce a cysteine mutation at the DFG-1 position in endogenously expressed FES kinase, for which complementary, covalent inhibitors were synthesized. Engagement of mutant FES by its cognate inhibitor allows for temporally controlled modulation of catalytic activity, thereby enabling the minimally-invasive study (previous reports rely on transient/stable overexpression or knock-out models) of its physiological role. Acute inactivation of mutant FES revealed two insights in HL-60 cells: 1) FES activity is dispensable during differentiation along the macrophage lineage and 2) SYK is a downstream target of FES, for which its phosphorylation regulates bacterial phagocytosis in mature neutrophils. Furthermore, mutant-specific labeling was achieved with three other kinases – LYN, PTK2, and PAK4 – after introducing a DFG-1 cysteine, suggesting the broad amenability of this position as a handle for covalent probe engagement.

Although the strategy employed in the present study is not entirely novel (mutagenic introduction of a cysteine to enhance potency and selectivity of an inhibitor has been used to study various other kinases: Poulikakos, 2010; Kung, 2017; etc.), this study successfully applies the chemical-genetic approach with endogenous levels of the kinase-of-interest and identified a new site amenable such a strategy. The importance of such a study is quite significant, especially considering recent reports detailing confounding results in overexpression and knock-out models. Moreover, extensive characterization of the mutants is performed, ensuring orthogonality of the cysteine mutation in HL-60 cell physiology. As such, this reviewer recommends conditional acceptance of the manuscript for publication in Nature Communications, contingent upon addressing several minor/moderate concerns.

Specific comments:

C1) SelectScreen™ screening of 279 kinases was performed to assess kinome-selectivity of WEL028 (supplementary figure 3). Based on the language in-text, it is slightly unclear what the selection criteria was for the kinases included in the screen. Do the 279 kinases contain “native cysteine residues at any position in the active site” (page 6) or is there a smaller sub-set of the 279 that contain such a cysteine? If the latter, why were these particular kinases selected for the screen and why were the others excluded?

R1) The selected panel consists of the 279 wild-type kinases that were available at this commercial platform at the time of the screening (in 2015). Not all of these kinases have native cysteines in the active site, and not all kinases with a native cysteine were available for testing at that moment.

In the revised version of the manuscript, we extended this single-dose selectivity screen with 101 additional kinases to provide a more complete and representative coverage of the entire kinome (Fig. 3e and Supplementary Fig. 4). Out of the total of 380 tested kinases, 354 kinases (93% of total) showed a >100-fold selectivity window against FES^{S700C} inhibition at 1 μM. Based on the new data, we were able to confirm that GAK is a target of WEL028 *in vitro*, in accordance with the results from our chemical proteomics experiments in live HL-60 neutrophils (Fig. 6c). We have added the data to the manuscript.

C2) Can the authors justify the need for including both a one-step (WEL033) and a two-step (WEL028) fluorescent probe in their study? Rather than incorporate the fluorescent tag directly on the inhibitor (and thus face cell permeability issues, fluorescent decay after extended storage,

etc.) the authors could easily perform the CuAAC reaction with Cy5-azide after covalent labeling of WEL028 on FES targets. Another alternative strategy to the target engagement competition experiments performed (in which pre-treatment with WEL028 is followed by treatment with WEL033) is to pre-treat cells, protein, etc. with inhibitor 5 and subsequently treat with WEL028, followed by Cy5-azide CuAAC. Is there a particular reason for performing one-step probe labeling experiments?

R2) We agree with the reviewer that fluorescent tags may affect the cell permeability of probes. However, in our studies WEL033 has only been used for *in vitro* labeling of lysates and not for cellular experiments. The use of one-step probe WEL033 allows fluorescent labeling of proteins in cell lysates in a single step, and does not require CuAAC as a separate experimental step. Consequently, the one-step labeling is more efficient for cell lysates.

Stock solutions of WEL033 were routinely stored protected from light as small volume aliquots and we never experienced fluorescent decay to be an issue in our experiments.

C3) Supplemental figure 5b appears to have mislabeled lanes. As labeled, pre-treatment with 4, followed by WEL028 treatment appears to lead to an intense fluorescent signal whereas WEL028 alone does not. Perhaps these samples were swapped during gel loading? Otherwise, the data seems to be irreconcilable with the rest of the data in the manuscript.

R3) Thank you for pointing this out. The lanes were unintentionally mislabeled. This has been corrected in the revised manuscript.

C4) In order to determine an irreversible, covalent mode of binding for WEL028, inhibitor washout experiments were performed (supplementary figure 6d,e). It is slightly concerning that the vehicle-treated sample post-dialysis leads to such a weak fluorescent signal. There is no obvious reason why post-dialysis intensity should be so much lower, despite lower protein levels in general. This is likely leading to an artificially low signal for WEL028-treated samples and an artificially high signal for TAE684-treated samples post-dialysis (considering normalization to vehicle). If the authors are satisfied with the quality of the data, how do they rationalize the relatively dramatic increase in fluorescence for TAE684-treated samples post-dialysis? Surely TAE684 isn't enhancing affinity for WEL033.

R4) We agree that this is a surprising phenomenon that was observed in all replicates (individual dialyses). The protein concentration was lower in post-dialysis than pre-dialysis samples, but the fluorescence intensities were corrected for relative protein loading and thus do not explain the high signal for TAE684-treated lysates. We hypothesize that TAE684 stabilizes the protein and reduces aggregation or (proteolytic) degradation. This phenomenon has previously been observed with other molecular chaperones as reviewed in Convertino, M. *et al.* ACS Chemical Biology **11**, 1471–1489 (2016).

C5) To ensure that FESS700C HL-60 cells differentiate in a manner identical to FESWT cells, proliferation and CD11b-receptor expression between the two alleles were monitored and compared (Fig 4e,f). The text contains mis-references (bottom of page 8 and top of page 9) to the figures: fig 4e contains cell proliferation data while fig 4f describes CD11b expression percentages.

R5) Our apologies, Fig. 4e and 4f have been reversed in the revised manuscript, so that the references in the main text are now correct.

C6) The authors provide very little discussion regarding the presence of several off-targets (Fig 4g) besides a brief acknowledgement of their existence (page 9). The kinome screen performed revealed a few kinases (LRRK2, ~286 kDa; MKNK2, 46 kDa) for which WEL028 is only roughly 10-fold less selective against. The chemical proteomic analysis revealed additional ones (MAPK1/MAP2K2/MAP2K1/GAK etc.). Under the conditions described (1 μ M WEL033), is there

any concern that the off-targets could potentially complicate downstream biological characterization? The authors should discuss this important point.

R6). The advantage of our chemical genetic approach is that we take the off-targets of WEL028 into account by also performing comparative, biological studies using wild-type cells. Thus, for the interpretation of the biological results, we do not solely rely on the selectivity profile of the compound, but also take also the genetic background of the cells into account. For example, we showed that the off-targets of WEL028 are responsible for the inhibition of macrophage differentiation in both wild-type cells and mutant cells, while the WEL028-induced inhibition of phagocytosis of *E. coli* in neutrophils is only observed in the mutant cell line. In the manuscript we have discussed that the off-targets MAP2K1/2 could be responsible for the inhibition of HL60 differentiation. In summary, the biological effects of the off-target effects of WEL028 become apparent in the wild-type cells, and therefore, there is no concern in the interpretation of the downstream biological effects of WEL028 in the mutant FES^{S700C} cell line, if these biological effects are only observed in the mutant line and not in wild-type cells.

C7) A targeted transcriptomics analysis was performed using TempO-Seq to ensure that FESS700C HL-60 cells exhibit few to no transcriptional deviations during macrophage differentiation (Fig 5a). The authors conclude that the mutation minimally disturbs gene expression based on the presence of only 7 identified transcripts that are altered in mutant FES cells. While this appears rational from a quantitative perspective – 7 out of ~21,000 total transcripts is a small, insignificant percentage (0.03%) – there is no characterization of the qualitative effect of these alterations, albeit few. It seems perfectly plausible that even if only 1 altered transcript is identified, its alteration could have significant effects beyond the transcriptome level.

R7) We agree with the reviewer that even a single altered transcript might have effects in functional studies. However, it seems unlikely that any of the transcripts were having significant effects in this case, because we demonstrated that the wild-type and mutant HL-60 cells behaved similarly in the processes examined in this study: macrophage differentiation and neutrophil phagocytosis. Furthermore, none of the altered transcripts (*PRG2*, *MMP14*, *MALAT1*, *STT3B*, *RPL3*, *B2M* and *GLO1*) have previously been reported to play a role in these processes. We have included a remark in the main text to address this concern.

Reviewer #2 (Remarks to the Author):

Van der Wel and colleagues have engineered a FES kinase mutant with residue Ser700 near the activation loop changed from Ser to Cys. They then modified the TAE684 small molecule FES inhibitor that had previously been co-crystallized with FES (Hellwig et al. 2012) to find derivatives that were more selective inhibitors of FES-S700C, relative to FES-WT. Optimized molecules were further modified to include an electrophilic group that would form a covalent bond through a Michael addition to the novel thiol at C700. Variant of this compound were generated with a fluorescent group to facilitate labelling FES-S700C or biotin to enable affinity purification.

C8) Biochemical analysis indicates these compounds effectively and differentially inhibit FES S700C, relative to FES WT. However, the specificity of these compounds with respect to the kinome is overstated as they still inhibited 50% of the activity of 19 of 279 kinases profiled at 1uM. By extension, one would expect ~38 of the human 518 kinases to be significantly inhibited at 1uM concentration. This degree of specificity is comparable with a wide range of existing kinase-“specific” inhibitors. Thiol directed covalently acting kinase inhibitors are not novel; and this set of inhibitors are not particularly specific for the FES-S700C target.

R8) To address the concern of the reviewer, we have adapted the selectivity claims with regard to the general specificity of the compound. It should be noted that the kinome screen was performed at 1 μ M, but 100 nM of WEL028 is sufficient to completely inactivate FES^{S700C} in living cells. At this lower concentration WEL028 has an improved selectivity profile (Fig. 6a,c), which is sufficient for the current

studies. We agree that more selective inhibitors are available for some specific kinases (e.g. members of the MEK family), but to our knowledge, selective FES inhibitors (e.g. no cross-reactivity with FER) with demonstrated cellular efficacy have not been reported so far.

C9) The rationale for developing this thiol-directed inhibitor for an engineered FES kinase was confusing because the authors discussed it in the context of ATP analog sensitive kinase derivatives used by Shokat and colleagues to identify kinase substrates. In the Shokat approach, it is the gatekeeper hinge region of the kinase that is altered to allow derivative forms of ATP to be used as substrates to selectively phosphorylate and thereby identify kinase substrates. In contrast, the work described here by Van der Wel and colleagues is directed at selectively inhibiting and covalently labeling an engineered FES variant.

R9) We apologize if the rationale for our work was confusing. Our primary aim was to develop a method to determine cellular target engagement of kinases. Target engagement is an essential step in drug discovery and helps to study the function of kinases and to validate kinases as drug targets by pharmacological tools. Target engagement correlates inhibitor exposure at the site of action to a pharmacological and phenotypic readout. To develop a target engagement method, we have employed a chemical genetics approach, which was inspired by Shokat and colleagues. They have previously (in addition the ATP analog-sensitive kinase derivatives) reported on the concept of 'covalent complementarity', directed at covalently inhibiting engineered kinases with a mutated gatekeeper residue (cysteine). The mutant kinases in these former studies, however, suffered from diminished kinase activity and required a secondary mutation to improve cysteine reactivity or compound selectivity and potency. We identified the DFG-1 residue as a promising position to mutate into a cysteine without affecting kinase function. In addition, we for the first time apply CRISPR/Cas9 gene editing to introduce the mutation endogenously in a cell line, whereas previous reports relied on (transient) overexpression of the mutant kinase. We have adjusted the manuscript to enhance the clarity of our primary goal.

C10) Unfortunately, the compounds they developed still inhibit and label many other kinases/targets. The extent to which these reagents are selective towards the engineered FES-S600C variant is overstated.

R10) We have adapted our claims about the selectivity of WEL028. See also our comments in R8.

C11) CRISPR/Cas9 methods were then used to engineer the same S700C substitution into the endogenous FES locus in HL60 cells, a human promyelocytic leukemia cell line that can be induced to differentiate into granulocytes or macrophages. The rationale for this was to be able to explore the effects of selectively inhibiting FES on differentiation, proliferation, bacterial phagocytosis and signaling in a physiologically relevant context. They make an argument that this will be more informative than a genetic approach because that is prone to adaptation. This argument is also overstated because cells can also adapt to drug challenges; but of more concern, their inhibitors may act on other kinases (or other targets) in this system whereas genetic approaches are more likely to be specific. Surprisingly, the authors did not generate a simple CRISPR/Cas9 FES knockout HL60 cell line and use it to perform a head to head comparison.

R11) As indicated in R9, the primary goal of our study was to develop a method to determine cellular target engagement of kinase inhibitors to correlate their biological effect with target occupancy (Fig. 1). A FES knockout model cannot be used to achieve this goal, therefore we did not perform a head-to-head comparison. We emphasize that our method is not intended to replace other genetic methods, such as knockout models, but serves to help target validation of kinases using pharmacological tools. For example, we could demonstrate that the differentiation of HL60 into macrophages or neutrophils is not dependent on inhibition of FES by WEL028 at 1 μ M, but on its off-targets. In contrast, phagocytosis of *E. coli* by neutrophils is mediated by FES, because WEL028 (at 100 nM, 1h) fully inhibited FES^{S700C} activity and reduced phagocytosis in the mutant FES^{S700C} cells, but not in wild type cells.

Indeed, prolonged drug exposure may cause desensitization, tolerance or degradation of some proteins, which may affect the ability of the protein to engage with a drug. With our method, it is possible to take some of these effects into account.

C12) CD11b expression in PMA-induced HL60 cells was used to assess myeloid differentiation. These data (Fig 5d,e) would seem to indicate that FES suppresses differentiation. However, statistics were not shown in the case of the data in Figure 5e, where WT and FES-S600C cells were compared.

R12) This is an interesting observation of the reviewer. The CD11b expression is indeed statistically significantly increased, but it is a very small effect (3% increase) despite complete FES inhibition (Fig. 5b). Of note, CD11b expression was not the only experimental evidence that we used to assess myeloid differentiation: we also observed no significant changes in proliferation of cells treated with vehicle or 100 nM WEL028 (Fig. 5e). This further supports that wild-type and mutant cells were differentiated to the same extent upon WEL028 incubation. Thus, we do not regard the small increase in CD11b expression to be biologically relevant, and attribute it to experimental variation. See also our comment in R17.

C13) The greater effect on myeloid differentiation at 1uM was interpreted as due to off-target effects. Indeed, the labeling experiments in Fig 4g and Fig 5b indicate that FES-S700C represents a minority of the proteins that are tagged with their inhibitor. The authors next show data indicating that FES-S700C inhibition with WEL028 can reduce bacterial phagocytosis. (Fig 6d,e). It was not clear how they quantified the data in Fig 5d to arrive at the phagocytic indices in the graphs shown in Fig 5e,i.

R13) A more detailed explanation of the calculation of phagocytic indices is included in the legend of Fig. 6d-e and Supplementary Fig. 15 as well as in the Supplementary Materials & Methods. In addition, we now included the raw and analyzed data in the corresponding Source Data file.

C14) In Fig 5e, where WT and FES-S600C cells are compared, the effect looks to be ~25%. This is consistent with the 36% reduction in bacterial phagocytosis that was previously reported in FES knockout mouse resident peritoneal macrophages relative to WT (Parsons and Greer, 2006. J Leukoc Biol. 80:1522). Van der Wel and colleagues cite this paper a couple times in their manuscript, but do not state that a role for FES in bacterial phagocytosis was previously demonstrated in that study. Other previous studies using mice targeted with either null or kinase-inactivating mutations in Fes have also demonstrated that FES is not required for hematopoiesis (Senis et al 1999. Mol Cell Biol 19:7436; Senis et al 2003. Exp Hematol. 31:673), including the generation of granulocytes and macrophages; and one of these studies also explored SYK, PLCg2 and HS1 tyrosine phosphorylation during platelet activation and saw no effects with FES genetic disruption (Senis et al 2003. J. Thrombosis and Haemostasis 5:1062).

R14) Thank you for alerting us to these relevant findings. It clearly demonstrates that FES has cell type-specific effects. Apparently, FES does not activate the SYK-pathway in platelets, whereas FES is involved in phagocytosis in neutrophils (current study) and in macrophages (Parsons and Greer). Of note, Parsons and Greer did not use pharmacological tools and did not identify SYK as a downstream mediator of FES. Thus, our studies are complementary and reinforce each other. It is tempting to speculate that the type of external stimulus (*e.g.* collagen vs. live bacteria) influences the downstream signaling pathways of FES. We have adapted the discussion accordingly on page 15.

C15) Van der Wel and colleagues show some evidence for a physical association between FES and SYK in a co-overexpression co-IP experiment (Fig 6h). Adding the reciprocal co-IP would have made this more compelling. They also show some evidence for FES mediated phosphorylation/activation of SYK; again in a co-overexpression system. SYK pY352 was abolished by WEL028, and this was interpreted as evidence for FES mediated SYK activation. However, in Sup Fig 11 it is apparent that this SYK pY352 signal is also seen in the absence of

co-transfected FES, so it seems very likely that SYK can autophosphorylate under these over-expression conditions. It is difficult to reconcile these two pieces of data (Fig 6g and Sup Fig 11). How is it that SYK pY352 is inhibited by WEL028 in the presence of co-overexpressed FES-S700C, but not in its absence?

R15) To clarify this point, we have performed additional experiments that indeed indicate that SYK is autophosphorylated in absence of FES (loss of pY352 when only kinase-dead SYK^{K402A} is expressed). This autophosphorylation is not sensitive to WEL028, which is in accordance with our observations that WEL028 does not target SYK (Fig. 6c and Supplementary Fig. 4). When SYK is co-expressed with FES, Y352 phosphorylation becomes completely dependent on FES activity. In this case also kinase-dead SYK^{K402A} shows phosphorylated Y352 and this phosphorylation is blocked upon incubation with FES^{S700C}-specific inhibitor WEL028. These new data are added in Supplementary Fig. 16b.

C16) They then explore signaling in HL60 FES-S700C cells that have apparently been differentiated into neutrophils (although it is was not specifically stated that these cells were treated with ATRA/DMSO).

R16) We apologize for the confusion and now specifically stated this in the figure legend.

C17) SYK, PLCg2 and HS1 phosphorylation was observed after incubation with bacteria, and this was inhibited by WEL028. Surprisingly, this experiment was not done on control HL60 cells in parallel. In the absence of this control, it is not valid to conclude that the loss of phosphorylation of SYK, PLCg2 and HS1 in the presence of WEL028 is due to FES inhibition.

R17) We thank the reviewer for raising this point. To this end, we have performed additional experiments in which we incubated both WT and FES^{S700C} HL-60 neutrophils with vehicle or 100 nM WEL028 (1 h), followed by stimulation with live *E. coli* (now Fig. 6j).

We found that SYK, PLCγ2 and HS1 phosphorylation is unaffected by WEL028 in WT neutrophils, whereas WEL028 blocks phosphorylation of these proteins in FES^{S700C} neutrophils. These results are in accordance with the flow cytometry assay (Fig. 6d-e), which showed no effect of WEL028 on phagocytic uptake by WT neutrophils. Together with the WEL028 target profiles on WT and FES^{S700C} neutrophils identified using chemical proteomics (Fig. 6c), these results collectively confirm that the loss of SYK, PLCγ2 and HS1 phosphorylation is due to specific, on-target inhibition of FES.

Importantly, to further support our conclusion that FES is involved in the activation of the SYK-HS1-PLCγ2 pathway with an independent genetic method, we also generated a FES knockout (FES^{KO}) HL-60 cell line using CRISPR/Cas9. Details about the generation and validation of this FES^{KO} cell line can be found in Supplementary Fig. 18 and Supplementary Materials & Methods – Biology. Notably, we observed that FES^{KO} cells maintained the ability to differentiate into neutrophils (Supplementary Fig. 18e), which is in accordance with our results showing that pharmacological FES inactivation does not affect neutrophil differentiation of HL-60 cells (Supplementary Fig. 12). These findings further supports our comments in R12. Importantly, FES^{KO} neutrophils displayed severely impaired SYK, HS1 and PLCγ2 phosphorylation in comparison with WT (and FES^{S700C}) neutrophils. These data thus further support the role of FES in SYK, HS1 and PLCγ2 phosphorylation in response to *E. coli* infection.

Please note that individual protein loading controls (anti-SYK, anti-PLCγ2 and anti-HS1 immunoblots) in Fig. 6j are missing due to acute closure of our laboratory following the COVID-19 pandemic. We do have a general loading control (β-actin) included, which showed that protein transfer was equal for all samples. We apologize for this omission, but we thank you for your understanding in these special circumstances. As discussed with the editor, the immunoblots will be incorporated as soon as our lab re-opens.

C18) Furthermore, they provide no evidence that FES is activated under these conditions of bacterial challenge.

R18) We respectfully disagree. By using WEL028 at 100 nM and the wild type cell line as control, we clearly show that bacterial phagocytosis is reduced by selective inhibition of FES (Figure 6a,c,e,j). From this observation we can conclude that FES is active. Unfortunately, the commercially available anti-phospho-FES Y713 antibody to detect activated FES worked only in overexpression systems and could not be used to detect endogenous FES (auto)phosphorylation.

C19) The final figure provides a model for FES recruitment to sites of bacterial phagocytosis and downstream activation of SYK and subsequent activation of PLC β 2 and HS1. This is a highly speculative model with very little evidence in the manuscript to support it.

R19) We respectfully disagree with the reviewer. We have provided a step-by-step explanation of the model which is supported by appropriate references to the data in our manuscript and to previous literature reports (including the paper of Parsons and Greer (2016) as mentioned by the reviewer; as well as a comprehensive review of Craig (2011) (ref. 11). In brief:

Fig. 7a is supported by references 19 and 61.

Fig. 7b is supported by references 10 and 62.

Fig. 7c is supported by Fig. 6d-e, 6h, 6i-j and references 19, 63, 64 and 65.

Fig. 7d is supported by Supplementary Fig. 17 and reference 66.

Since the model is explained in detail in the discussion section of our manuscript, we have changed Fig. 6k into Fig. 7 (in the discussion).

C20) Other points: The PamChip peptide array-based comparison of FES-WT vs FES-S700C activity looks impressive (Fig 2e), but it would be more compelling if a different kinase was included as a control. How many of the peptides on this array are validated FES targets? The same issue applies to the SH2 binding PamChip analysis (Fig 2g); and in this case, there do appear to be some significant differences that the authors don't acknowledge.

R20) We have now included PamChip substrate profiles of five other kinases in the non-receptor tyrosine kinase family: ABL, CSK, FGR, LYN and SYK (Supplementary Fig. 2). A substantial number of the identified peptide FES substrates are not phosphorylated by other structurally similar kinases in this assay.

Some of the peptides identified as FES substrates in the PamChip array were previously validated in literature, such as the PECA1 (Tyr713, Udell, C. M. *et al.* J. Biol. Chem. **281**, 20949–20957 (2006)) and FES itself (Tyr713, autophosphorylation, Rogers, J. A. *et al.* J. Biol. Chem. **271**, 17519–17525 (1996)). Relatively few validated FES substrates have been reported to date, and not all known FES substrates are covered by the peptides on the microarray. Overall, the consensus sequence identified in this study (Fig. 2f) is in line with a previous report reporting on FES substrate recognition (Filippakopoulos, P. *et al.* Cell **134**, 793–803 (2008)).

The SH2 binding PamChip array was specifically developed for this study, and this assay requires a protein construct with both a functional SH2 and kinase domain. Unfortunately, no comparative studies with other kinases are currently available. It is outside the scope of the current manuscript to generate these data for other SH2-kinase constructs. We do realize that the absolute signal intensities in the SH2 binding experiment are consistently higher for FES^{S700C} compared to FES^{WT}. It should be noted, though, that the peptide order ranked by signal intensity is very similar for both proteins. The slight differences in absolute signal intensity may possibly be explained by a minor difference in protein input or by a minimal variation in the efficiency of the peptide pre-phosphorylation step. It should be noted that this microarray assay is very sensitive to these factors, as the signal output is directly proportional to the

amount of protein bound to the phosphorylated peptide. For these reasons, we only used this SH2 binding profile as a qualitative assay to compare the binding profiles of FES^{WT} and FES^{S700C}.

C21) In Figure 4g the presumptive 93 kDa FES band is one of several species labeled with 1 μ M WEL033 in HL60 cells. FES is a minor species, likely in the range of <10% of the labeled proteins. This speaks to the lack of specificity of this probe.

R21) WEL033 is used at a concentration of 1 μ M *in vitro* to effectively detect protein targets by SDS/PAGE, whereas 100 nM of WEL028 is sufficient to completely inactivate FES in live cells (*in situ*). At 100 nM, WEL028 is more selective as shown by the competitive labeling experiment (Fig. 6a) and chemical proteomics experiments under these conditions (Fig. 6c). The combination of the mutant cell line with the complementary inhibitor and the wild-type control cell line provides the required specificity. We have adapted our claims about the selectivity of the probe. See also our comments in R8.

Reviewer #3 (Remarks to the Author):

This study reports the development of a chemical genetics strategy to probe the role of the c-Fes non-receptor tyrosine kinase in macrophage differentiation and neutrophil function. The approach involves a single amino acid substitution of Ser700 with Cys in the kinase domain, which in turn enables irreversible inhibition with an electrophilic compound based on TAE844, a competitive inhibitor previously reported to inhibit Fes. The mutant kinase is characterized extensively *in vitro*, and then engineered into the genome of the human HL-60 promyelocytic leukemia cell line using CRISPR technology to avoid artefacts related to over-expression. Using the modified cells and probe compound, the authors show that Fes activity is not required for induction of macrophages from HL-60 cells following phorbol ester treatment. However, they do show a role for Fes activity in phagocytosis of bacteria by neutrophils derived from HL-60s, through a pathway linked to activation of Syk.

Overall this is a very impressive study that effectively combines many aspects of chemical biology and mass spectrometry in support of the ultimate conclusions. The manuscript is very well written, and the data are presented in a clear and logical manner. Below are a few questions and comments for the authors to consider:

C22) In the Introduction (page 2), the authors state that Fes knock-out mice revealed a role for Fes in myeloid differentiation. In fact, the opposite is true - Fes homozygous-null animals did not show any defect in myelopoiesis.

R22) We apologize for this inconsistency and adjusted this phrase accordingly.

C23) Results, page 5: In the text and in Table 1, please mention the concentration of ATP used for the kinase inhibition assays, and the relationship of the ATP concentration to the K_M for each kinase

R23) The K_M of both kinases is shown in Fig. 2d and mentioned in the main text on page 4; we now also included the used ATP concentration in the main text on page 5. In addition, we included these details in the legend of Table 1 and in Supplementary Fig. 1

C24) Regarding the experiments with HL-60 cells, it would be very helpful to the reader to include in the main text and the figure legends more details of the differentiation protocols. Please specify the agents used to induce differentiation to macrophages vs. neutrophils along with the concentrations used and the period of time required for differentiation induction.

R24) We now specified the differentiation agents, concentrations and incubation times required for macrophage and neutrophil differentiation in the main text (page 8) and legend of Fig. 4a.

C25) Also no phenotypic or morphological data is included to verify the extent of neutrophil differentiation in response to ATRA or DMSO.

R25) In the revised manuscript, we added additional phenotypic evidence for differentiation into functional neutrophils with the use of a respiratory burst assay (Supplementary Fig. 12e-f). In this assay, cells are incubated with nitroblue tetrazolium (NBT) prior to stimulation with PMA to induce a respiratory burst, due to which NBT will undergo reduction to form an insoluble blue-black product: formazan. Microscopy analysis revealed that 70-80% of the cells after neutrophil differentiation were capable of inducing a respiratory burst. Notably, acute inhibition of FES^{S700C} with WEL028 prior to PMA stimulation did not affect the percentage of formazan-positive cells.

C26) In Figure 4, panels e and f should be reversed.

R26) Our apologies, Fig. 4e and 4f have been reversed in the revised manuscript. See also R5.

C27) In Figure 5b, it is clear that treatment with WEL028 selectively engages Fes S700C vs. WT Fes. However, does this correlate with complete suppression of Fes kinase activity? This could be confirmed by independent IP of Fes followed by immunoblotting with the pY713 antibody.

R27) Unfortunately, the commercially available anti-phospho-FES Y713 antibody to detect activated FES worked only in overexpression systems. It should be noted that FES target engagement consistently correlated with inhibition of FES kinase activity in all biochemical and cellular (co-expression) experiments in our study. At 100 nM WEL028, FES activity was completely inhibited as evidenced by recombinant kinase activity assays (Fig. 3b), PamChip microarray assays (Fig. 3c and 6f), probe labeling assays using cell lysates (Fig. 3i-l) and live cells (Fig. 5a-b, 6a-b), analysis of FES autophosphorylation in overexpression systems (Fig. 3m-n) and phosphorylation of downstream proteins in neutrophils (Fig. 6j). From these observations we conclude that FES target engagement correlated with FES activity; and that FES activity was completely suppressed by WEL028 in the stimulated neutrophils.

C28) Figure 7h shows a very powerful control, in which kinases labeled with WEL028 at a high concentration (1,000 nM) are captured and identified by MS to define the off-targets. Has this experiment been performed at 100 nM? Such a result, showing that only Fes is captured at the lower concentration, would provide very strong evidence for on-target performance of this compound at an appropriate concentration.

R28) We have performed a chemical proteomics experiment at 100 nM (Fig. 6c), where FES, GAK and MAPK1 were identified as the targets of WEL028 under these conditions. Notably, the GAK and MAPK1 were also detected in the wild-type controls (Supplementary Fig. 14).

C29) Data presented suggest that FES activity is not necessary for PMA-induced differentiation of HL-60 cells to macrophages. Did the authors test whether HL-60 differentiation along the neutrophil pathway via all-trans retinoic acid or DMSO treatment also requires FES activity? Early studies established that Fes activity is induced by these agents, but whether its activity is required for differentiation is not clear.

R29) To address this question, we performed additional experiments of which the results are presented in Supplementary Fig. 12 of the revised manuscript. We found that FES activity is not required for ATRA/DMSO-induced differentiation of HL-60 cells to neutrophils, similar to our observations for PMA-induced differentiation to macrophages. Notably, we did observe an increase in FES expression in HL-60 neutrophils compared to non-differentiated cells (Fig. 4h), which could explain why earlier studies observed an increase in FES activity in response to ATRA and/or DMSO.

Reviewer #4 (Remarks to the Author):

In the manuscript “Chemical genetics strategy to profile kinase target engagement reveals role of FES in neutrophil phagocytosis via SYK activation” from the van der Stelt lab, the authors designed a mutant specific probe for the FES kinase and used these tools to understand the role of this kinase in downstream signaling pathways including myeloid differentiation and endocytosis.

The paper is interesting. By introducing a Cys in close proximity to highly conserved DFG motif, they could ablate the function the FES kinase with covalent interaction of probe and further investigated the downstream signaling pathway. Using competition assays, rational design and biochemical assays the authors demonstrated the specific interaction of probe with FES kinase. Moreover, by acute inhibition of FES, they show that FES is dispensable for differentiation of myeloid lineage; instead, they show this kinase has a role in receptor mediated endocytosis of *E. coli*. They showed that in this process the FES protein become auto phosphorylated and induced phosphorylation of other downstream kinase such as PLC gamma2 and HS1. Overall, the paper is rational and tried to use both genetic and chemoproteomic approaches to understand the role of FES as an important kinase involved in cancer and immune disorders.

Major point:

Here, the author tries to map the probe binding site and see that besides FES, several other proteins can be inhibited with the probe (the probe is not specific for only FES kinase). As shown in the data, the probe can interact with other cysteines close to the DFG motif. This results in a lot of complexity to the signaling pathway downstream of their specific inhibitor and understanding the function of FES. The result shows convincing data that the role of FES is not important in the process of myeloid differentiation.

C30) As next step, the role of FES kinase has been studied in the context of endocytosis process and phosphorylation of several targets shown to be disrupted upon probe treatment. Here, a critical control is missing due to the fact that this might be due to the off-target activity of the probe. The following should be addressed: The authors should include both control and CRISPR modified cell line treated with inhibitor (with different concentrations to establish dose dependence) to see whether the signalling pathway can be regulated differently (the phosphorylation events upon *E. coli* infection. Furthermore, the study of signalling pathways in a FES knockout cell line can be considered as further support of the role of FES kinase in endocytosis.

R30) We thank the reviewer for these valuable suggestions.

Concentration-response experiments have been performed to determine which WEL028 concentration was required for complete FES inhibition (Fig. 3k-l, 5a-b, 6a-b and Supplementary Fig. 13). This information about FES target engagement was instrumental to select a concentration of 100 nM WEL028 for acute FES inactivation in the neutrophil phagocytosis experiments.

As requested by the reviewer, we have performed additional experiments and included the wild-type HL-60 neutrophils as a control in the stimulation experiments with *E. coli* (now Fig. 6j). These data show that inhibition of SYK, HS1 and PLC γ 2 phosphorylation by WEL028 is only observed in FES^{S700C} neutrophils and not in WT neutrophils, which confirms an on-target, FES-specific effect. See also R17.

Importantly, to further support our conclusion that FES is involved in endocytosis with an independent genetic method, we also generated a FES knockout (FES^{KO}) HL-60 cell line using CRISPR/Cas9. Details about the generation and validation of this FES^{KO} cell line can be found in Supplementary Fig. 18 and Supplementary Materials & Methods – Biology.

Notably, we observed that FES^{KO} cells maintained the ability to differentiate into neutrophils (Supplementary Fig. 18e), which is in accordance with our results showing that pharmacological FES inactivation does not affect neutrophil differentiation of HL-60 cells (Supplementary Fig. 12).

Interestingly, FES^{KO} neutrophils displayed severely impaired SYK, HS1 and PLC γ 2 phosphorylation in comparison with WT (and FES^{S700C}) neutrophils. These data thus further support the role of FES in SYK, HS1 and PLC γ 2 phosphorylation in response to *E. coli* infection.

Please note that individual protein loading controls (anti-SYK, anti-PLC γ 2 and anti-HS1 immunoblots) in Fig. 6j are missing due to acute closure of our laboratory following the COVID-19 pandemic. We do have a general loading control (β -actin) included, which showed that protein transfer was equal for all samples. We apologize for this omission, but we thank you for your understanding in these special circumstances. As discussed with the editor, the immunoblots will be incorporated as soon as our lab re-opens.

C31) Moreover, it would be interesting to see whether the mutation in tyrosine 713 in FES kinase or dominant negative form of the kinase lead to the same defect in the signalling pathway.

R31) Thank you for this interesting suggestion. To investigate whether SYK Y352 phosphorylation is abolished downstream of the dominant negative, kinase-dead FES^{K590E} variant in a similar fashion as FES^{S700C} treatment with WEL028, we performed additional experiments (Supplementary Fig. 16a). Indeed, SYK Y352 phosphorylation is lost upon co-expression with kinase-dead FES^{K590E}. Genetic inactivation of FES thus phenocopies the effects FES^{S700C} treatment with WEL028.

REVIEWERS' COMMENTS:

Reviewer #1 (Remarks to the Author):

The authors have addressed this reviewer's comments satisfactorily.

Reviewer #2 (Remarks to the Author):

Review of van der Wel et al.

Comments to the authors:

The authors have made extensive revisions to this manuscript that greatly improve it. In response to two reviewers, they have generated an HL60 FES knockout line as a control. This is particularly important in the experiments with HL60-derived neutrophils challenged with bacteria. The absence of phosphorylation of SYK, PLC γ and HS1 (Fig 6j) upon bacterial challenge in these FES knockout cells, in addition to including the wild type cells was essential to support the claim that this SYK activation was FES-dependent. It would be important to finish this experiment with blots showing that comparable levels of these proteins are present in the lysates. Also, the legend to this figure does not even mention the FES KO cells; so that needs to be addressed. I congratulate the authors on this substantial improvement to the paper, which largely addresses a major concern; however, I would still point out that they have not demonstrated in this endogenous system (rather than the overexpression models) that FES is activated, or that it directly phosphorylates SYK during bacteria phagocytosis in these neutrophils. All of the co-overexpression experiments are consistent with that interpretation, but it must be acknowledged that these experiments don't prove FES activation or direct SYK phosphorylation in the physiologic context. There are still other possibilities, including an indirect role for FES in activating another kinase that is responsible for SYK activation.

I had previously asked about how the phagocytosis data was quantified. The raw flow cytometry data tracings of the bacteria phagocytosis assays are not compelling (Fig 6d), and the graphical quantitation show only a modest effect in Fig 6e, but apparently a larger effect in Fig 6i. Now that they have FES KO HL60 cells, one wonders if they have performed phagocytosis assays with those cells. It would be much more compelling if they could show that inhibitor treatment of the S700C cells phenocopied the phagocytosis defect in KO cells, and inhibitor treatment of the KO cells did not

further suppress their phagocytosis activity. This more fulsome analysis would compensate for the lack of novelty regarding the actual FES phenotypes described in this manuscript, since the lack of FES requirement for myeloid differentiation and role for FES in bacterial phagocytosis has been previously documented. This would provide readers with a comparison of what these authors have done with the usual approach of genetically knocking out the target and showing that inhibitor treatment of wild type cells phenocopies the knockout cells, without having any spurious (off-target) effects on the knockout cells.

On page 2 (lines 68-70) the authors state: "...phenotypic differences between independently generated knock-out animals are not uncommon, as was the case for two independently developed *fes*^{-/-} mice (17,26)." The authors should be aware that the Hackenmiller 2000 paper (#26) did indeed claim a defect in the myeloid lineage in their *fes*^{-/-} mice; but this was not seen in either the null (knockout, *fes*^{-/-} mice reported in Zirngibl 2002 (#17) or in the first report of a *fes* targeted mouse, a kinase-dead knockin mouse model (Senis, 1999 - PMID:10523632). These three papers used three different *fes* targeting approaches; and the phenotypic analysis in two of them showed no effect on myeloid differentiation (Senis, 1999 and Zirngibl, 2002). In the Zirngibl 2002 report, the authors point out a flaw in the targeting strategy used in the Hackenmiller 2000 paper, which could have disrupted expression of the closely linked *FURIN* gene. *FURIN* encodes a protease that is important for regulation of numerous targets (NOTCH, TGF β , NGF, vWF, MT-MMP) including some that are involved in hematopoiesis. So, while I would agree with the authors premise that there can be phenotypic differences with independently generated KO mice, it is important to delve into the details to understand why. In this case, I would agree with Reviewer #3, and suggest that a careful reading of the FES transgenic mouse literature shows that loss of FES does not affect myeloid differentiation.

Minor comments:

Figure 3 shows a Coomassie stain; but this is not a suitable control for how much recombinant FES is expressed. It needs to be a blot for FES or the Flag tag on the recombinant protein; as they show in Sup Fig 6b.

As a matter of curiosity, it would be helpful to the readers if the authors stated how many HL60 clones they analyzed to find the one S700C targeted clone.

Reviewer #3 (Remarks to the Author):

The authors have carefully considered and addressed all of my comments to the original version of their paper. Substantial additional work has performed in response to all four reviews, which has strengthened the work and the conclusions.

Reviewer #4 (Remarks to the Author):

The authors have provided additional data to support their claims. They use different strategies including both chemoproteomics together with genetic editing to demonstrate the power of a multifaceted approach in answering the role of FES kinase in the biology of myeloid differentiation and neutrophils. By targeting a cysteine in close proximity of the active site, which does not affect the biology of FES kinase, they demonstrated FES is essential in the process of phagocytosis and furthermore that the enzyme is dispensable for the process of myeloid differentiation. Their data showed that treatment with an inhibitor at low concentrations in comparison with high dose (due to the presence of cys) can demonstrate the role of FES kinase in this physiological process.

minor points:

- Line 485: We replaced one single atom out of 13.171 atoms in a protein by changing 485 only one base-pair out of over 3.000.000.000 base-pairs in the human genome

I think here the sentence should be changed due to the fact that off-target activity is always a possibility with CRISPR. Also, the format here is odd. Can the authors simply state 3 billion base pairs?

- The titles of results section introduced by step number (e.g. "Step 1 - Biochemical characterization of engineered FES kinases") is a bit odd. Authors should consider changing the format.
- The discussion section is unusually long. While there is certainly many findings to discuss, the authors should consider consolidating the discussion to be more succinct and to the point. Also, a paragraph describing some of the limitations of this approach could be appropriate so readers can decide whether this strategy is appropriate for their system. For example, this method is most appropriate when a crystal structure of the target kinase, availability of a non-catalytic cysteine, and a matching cysteine-directed covalent probe are available and likely not generalizable across the kinome.

Reply to reviewers for manuscript “Chemical genetics strategy to profile kinase target engagement reveals role of FES in neutrophil phagocytosis” (NCOMMS-19-38211).

Please find below a point-by-point reply (R) to the reviewers’ comments (C), followed by any revisions that were made in the manuscripts. We appreciate the time invested by the reviewers in evaluating our manuscript and thank them for their valuable comments and insights.

Reviewer #1 (Remarks to the Author):

C1) The authors have addressed this reviewer’s comments satisfactorily.

R1) We thank the reviewer.

Reviewer #2 (Remarks to the Author):

Review of van der Wel et al. Comments to the authors:

C2) The authors have made extensive revisions to this manuscript that greatly improve it. In response to two reviewers, they have generated an HL60 FES knockout line as a control. This is particularly important in the experiments with HL60-derived neutrophils challenged with bacteria. The absence of phosphorylation of SYK, PLC γ and HS1 (Fig 6j) upon bacterial challenge in these FES knockout cells, in addition to including the wild type cells was essential to support the claim that this SYK activation was FES-dependent. It would be important to finish this experiment with blots showing that comparable levels of these proteins are present in the lysates.

R3) We thank the reviewer for the useful suggestions during the previous round of revisions. We finished the control experiments and now provide remaining immunoblots that show comparable levels of SYK, PLC γ 2 and HS1 in the lysates (now Fig. 7j).

C3) Also, the legend to this figure does not even mention the FES KO cells; so that needs to be addressed.

R3) We apologize for the missing information in the figure legend and thank the reviewer for pointing this out. This has been addressed in the revised manuscript.

C4) I congratulate the authors on this substantial improvement to the paper, which largely addresses a major concern; however, I would still point out that they have not demonstrated in this endogenous system (rather than the overexpression models) that FES is activated, or that it directly phosphorylates SYK during bacteria phagocytosis in these neutrophils. All of the co-overexpression experiments are consistent with that interpretation, but it must be acknowledged that these experiments don’t prove FES activation or direct SYK phosphorylation in the physiologic context. There are still other possibilities, including an indirect role for FES in activating another kinase that is responsible for SYK activation.

R4) As discussed previously, the commercially available anti-phospho-FES Y713 antibody to detect activated FES worked only in overexpression systems. Therefore, we could not, unfortunately, show FES autophosphorylation in the neutrophils. However, it should be noted that FES target engagement consistently correlated with inhibition of FES kinase activity in all biochemical and cellular (co-expression) experiments in our study. At 100 nM WEL028, FES activity was completely inhibited as evidenced by recombinant kinase activity assays, PamChip microarray assays, probe labeling assays using cell lysates and live cells, and phosphorylation of downstream proteins in neutrophils. From these

observations we conclude that FES target engagement correlated with FES activity; and that FES activity was completely suppressed by WEL028 in the stimulated neutrophils.

Although our overexpression data do suggest there is a direct interaction between FES and SYK, we do agree with the reviewer that we cannot exclude the possibility that FES-induced SYK phosphorylation in the neutrophils is mediated via another kinase. Therefore, we have toned down our biological claims in the discussion and removed part of the discussion to the supplementary information (as Supplementary Discussion).

C5) I had previously asked about how the phagocytosis data was quantified. The raw flow cytometry data tracings of the bacteria phagocytosis assays are not compelling (Fig 6d), and the graphical quantitation show only a modest effect in Fig 6e, but apparently a larger effect in Fig 6i. Now that they have FES KO HL60 cells, one wonders if they have performed phagocytosis assays with those cells. It would be much more compelling if they could show that inhibitor treatment of the S700C cells phenocopied the phagocytosis defect in KO cells, and inhibitor treatment of the KO cells did not further suppress their phagocytosis activity. This more fulsome analysis would compensate for the lack of novelty regarding the actual FES phenotypes described in this manuscript, since the lack of FES requirement for myeloid differentiation and role for FES in bacterial phagocytosis has been previously documented. This would provide readers with a comparison of what these authors have done with the usual approach of genetically knocking out the target and showing that inhibitor treatment of wild type cells phenocopies the knockout cells, without having any spurious (off-target) effects on the knockout cells.

R5) The main goal of our manuscript was to develop a novel method for target engagement studies of endogenously expressed kinases. The requested additional experiments, which were not raised previously, are outside the scope of our manuscript. We respectfully disagree with the comment about the lack of novelty. The biological role of FES in bacterial phagocytosis in neutrophils and the role of FES-mediated SYK phosphorylation has never been described before.

C6) On page 2 (lines 68-70) the authors state: "...phenotypic differences between independently generated knock-out animals are not uncommon, as was the case for two independently developed *fes*^{-/-} mice (17,26)." The authors should be aware that the Hackenmiller 2000 paper (#26) did indeed claim a defect in the myeloid lineage in their *fes*^{-/-} mice; but this was not seen in either the null (knockout, *fes*^{-/-} mice reported in Zirngibl 2002 (#17) or in the first report of a *fes* targeted mouse, a kinase-dead knockin mouse model (Senis, 1999 - PMID:10523632). These three papers used three different *fes* targeting approaches; and the phenotypic analysis in two of them showed no effect on myeloid differentiation (Senis, 1999 and Zirngibl, 2002). In the Zirngibl 2002 report, the authors point out a flaw in the targeting strategy used in the Hackenmiller 2000 paper, which could have disrupted expression of the closely linked *FURIN* gene. *FURIN* encodes a protease that is important for regulation of numerous targets (NOTCH, TGF β , NGF, vWF, MT-MMP) including some that are involved in hematopoiesis. So, while I would agree with the authors premise that there can be phenotypic differences with independently generated KO mice, it is important to delve into the details to understand why. In this case, I would agree with Reviewer #3, and suggest that a careful reading of the FES transgenic mouse literature shows that loss of FES does not affect myeloid differentiation.

R6) Indeed, our results are in line with two studies reported by Senis 1999 and Zirngibl 2002. These references were already included and discussed in the manuscript. We added an additional sentence in the discussion to further emphasize this.

Minor comments:

C7) Figure 3 shows a Coomassie stain; but this is not a suitable control for how much recombinant FES is expressed. It needs to be a blot for FES or the Flag tag on the recombinant protein; as they show in Sup Fig 6b.

R7) We included an anti-FLAG immunoblot along with a β -actin loading control to verify the expression of FES^{WT} and FES^{S700C}. The corresponding Fig. 3g (now Fig. 4b) has been adjusted accordingly. Uncropped gel and blot images can be found in the Source Data file.

C8) As a matter of curiosity, it would be helpful to the readers if the authors stated how many HL60 clones they analyzed to find the one S700C targeted clone.

R8) We have added a remark about the number of screened clones in the results section of the revised manuscript.

Reviewer #3 (Remarks to the Author):

C9) The authors have carefully considered and addressed all of my comments to the original version of their paper. Substantial additional work has performed in response to all four reviews, which has strengthened the work and the conclusions.

R9) We thank the reviewer.

Reviewer #4 (Remarks to the Author):

C10) The authors have provided additional data to support their claims. They use different strategies including both chemoproteomics together with genetic editing to demonstrate the power of a multifaceted approach in answering the role of FES kinase in the biology of myeloid differentiation and neutrophils. By targeting a cysteine in close proximity of the active site, which does not affect the biology of FES kinase, they demonstrated FES is essential in the process of phagocytosis and furthermore that the enzyme is dispensable for the process of myeloid differentiation. Their data showed that treatment with an inhibitor at low concentrations in comparison with high dose (due to the presence of cys) can demonstrate the role of FES kinase in this physiological process.

R10) We thank the reviewer.

minor points:

C11) Line 485: We replaced one single atom out of 13.171 atoms in a protein by changing 485 only one base-pair out of over 3.000.000.000 base-pairs in the human genome I think here the sentence should be changed due to the fact that off-target activity is always a possibility with CRISPR. Also, the format here is odd. Can the authors simply state 3 billion base pairs?

R11) We have rephrased this sentence in the revised manuscript.

C12) The titles of results section introduced by step number (e.g. "Step 1 - Biochemical characterization of engineered FES kinases") is a bit odd. Authors should consider changing the format.

R12) The step numbers have been removed and the subheadings have been adjusted accordingly.

C13) The discussion section is unusually long. While there is certainly many findings to discuss, the authors should consider consolidating the discussion to be more succinct and to the point. Also, a paragraph describing some of the limitations of this approach could be appropriate so readers can decide whether this strategy is appropriate for their system. For example, this method is most appropriate when a crystal structure of the target kinase, availability of a non-catalytic cysteine, and a matching cysteine-directed covalent probe are available and likely not generalizable across the kinome.

R13) We thank the reviewer for these suggestions. We have consolidated the discussion and relocated the sections about potential biological implications to the Supplementary Information (as Supplementary Discussion). In addition, we have added a paragraph describing limitations of our strategy that should help readers in their decision whether our strategy could be applicable to their work.